# Squamate reptiles challenge paradigms of genomic repeat element evolution set by birds and mammals

Giulia I.M. Pasquesi[1], Richard H. Adams[1], Daren C. Card [1], Drew R. Schield[1], Andrew B. Corbin[1], Blair W. Perry[1], Jacobo Reyes-Velasco[1,2], Robert P. Ruggiero[2], Michael W. Vandewege[3], Jonathan A. Shortt[4] & Todd A. Castoe[1]

Broad paradigms of vertebrate genomic repeat element evolution have been largely shaped by analyses of mammalian and avian genomes. Here, based on analyses of genomes sequenced from over 60 squamate reptiles (lizards and snakes), we show that patterns of genomic repeat landscape evolution in squamates challenge such paradigms. Despite low variance in genome size, squamate genomes exhibit surprisingly high variation among species in abundance (ca. 25–73% of the genome) and composition of identifiable repeat elements. We also demonstrate that snake genomes have experienced microsatellite seeding by transposable elements at a scale unparalleled among eukaryotes, leading to some snake genomes containing the highest microsatellite content of any known eukaryote. Our analyses of transposable element evolution across squamates also suggest that lineage-specific variation in mechanisms of transposable element activity and silencing, rather than variation in species-specific demography, may play a dominant role in driving variation in repeat element landscapes across squamate phylogeny.

[1] Department of Biology, University of Texas at Arlington, 501S. Nedderman Drive, Arlington, TX 76019, USA. [2] Department of Biology, New York University Abu Dhabi, Saadiyat Island, United Arab Emirates. [3] Department of Biology, Institute for Genomics and Evolutionary Medicine, Temple University, Philadelphia, PA 19122, USA. [4] Department of Biochemistry and Molecular Genetics, University of Colorado School of Medicine, Aurora, CO 80045, USA. Correspondence and requests for materials should be addressed to T.A.C. (email: todd.castoe@uta.edu)

Transposable elements (TEs) and other repetitive sequences represent a major fraction of vertebrate genomes—in most mammals, repeat elements comprise 28–58% of the genome[1,2], and may comprise more than two-thirds of the human genome[3]. Several decades of genome research has led to the prevailing view that genome size and genome repeat content are tightly linked, such that shifts in genomic repeat content are expected to result in proportional shifts in vertebrate genome sizes[4–6]. Recently, this correlation has come into question in favor of alternative hypotheses, such as the "accordion" model of co-variation between genomic DNA gained by repeat element expansion and genomic DNA lost through deletion[7]. It has also been demonstrated that the relationship between genome size and repeat content may vary between vertebrate lineages[4,5,8], with some lineages adhering more or less to a particular model or pattern[4,6,7,9], underscoring the value of comparative analyses across diverse lineages.

Within vertebrates, our understanding of genome and repeat element evolution is largely biased towards mammals and archosaurian reptiles (mainly birds). The emerging pattern from studies of these groups is that large differences in the repeat element landscape exist among major amniote vertebrate lineages, yet fairly little variation in repeat content and diversity are observed within major amniote groups. For example, estimates based on de novo annotation of TEs in mammal and bird species suggest 1.7-fold and 2.2-fold variation in TE content across species for each group, respectively[1,7]. Although squamate reptiles (lizards and snakes) represent a major portion of the amniote tree with over 10,000 species spanning more than 200 million years of evolution[10], variation in genomic repeat content across squamate reptiles has remained poorly studied. From the few studies to date, genome size appears to be highly conserved in squamate reptiles[11], yet the little that we know about repeat element variation suggests that squamate reptile genomes vary greatly in repeat element content[12,13].

Motivated to assess whether squamate reptile genomic repeat element landscapes adhere to patterns observed in birds and mammals, we analyzed genomic repeat landscapes across 66 squamate species using low-coverage random whole-genome shotgun sample sequencing data[12,13] and draft genome assemblies. We find that squamate reptile genomes indeed challenge the paradigm that genome size and repeat content are tightly linked, and the view that major differences in repeat element content occur only between lineages of amniotes. In addition to contributions from TEs, snake genome repeat content variation is further increased by the largest known instance of microsatellite seeding by long interspersed nuclear elements (LINEs) observed in any living organism. We also find evidence that multiple independent horizontal transfer events and highly idiosyncratic patterns of TE proliferation across squamates have further contributed to extreme variation in genome repeat content in this lineage. We further tested a demographic explanation for variation in repeat content, whereby fluctuations in the effective population size ($N_e$) of species impact the efficacy of selection against repetitive element insertion[14]. We find no evidence that $N_e$ explains the distribution and variation in characteristics of the repeat landscape in squamate reptiles, which indicates instead that variation in molecular mechanisms of TE proliferation, silencing, removal, and truncation may underlie the extreme repeat variation observed across squamates. Collectively, our findings challenge existing views related to repeat element and genome size co-evolution, and provide new evidence for unappreciated variation in genomic repeat content within and among major amniote lineages.

## Results

### Comparison of sampled and assembled genome data.
Our analyses of genomic repeat content were based on the assemblies of 12 squamate genomes (including 1 new and 11 published assemblies), and low-coverage, unassembled genomic shotgun read datasets obtained from 54 squamate species (Supplementary Data 1; Castoe et al.[13]). Previous studies have shown that genomic repeat content estimated from unassembled shotgun genomic datasets are similar to estimates derived from assembled genomes[12,13]. We confirmed this by comparing repeat annotations from assembled and unassembled genome data from the same species (Supplementary Fig. 1), and also confirmed that repeat estimates derived from unassembled genomic shotgun datasets are effectively independent of the amount of sequence data obtained (Supplementary Fig. 1).

### Genome size and repeat content in major amniote groups.
Squamate reptile genomes challenge the commonly accepted paradigm that genome size and repeat content are tightly linked[4–6], and also challenge the prevailing view that large variation in repeat content tends to be characteristic of major clades, rather than highly dynamic within clades[1] (Fig. 1). For example, mammalian genome sizes tend to be more highly variable (2.2–6.0 Gbp[11]; Supplementary Data 2) in comparison with squamate and bird genomes, yet genomic TE estimates demonstrate only moderate levels of clade-specific variation (33.4–56.3%, mean = 44.5%; Fig. 1a, Supplementary Data 3, and Supplementary Note 1). In contrast, birds have smaller genomes and higher conservation of genome sizes (1.0–2.1 Gbp[11]; Supplementary Data 2), with relatively low levels of TE content (4.6–10.4%, mean = 7.8%, with the only notable exception being the downy woodpecker with an extremely high genomic TE content of 22.5%, which we excluded as an outlier from analyses here; Fig. 1b, Supplementary Data 3, and Supplementary Note 1).

With highly conserved genome sizes (1.3–2.8 Gbp) yet extensive variation in genomic content of readily detectable TEs (23.7–56.3%, mean = 41.8%; Fig. 1c), we find that squamate reptiles do not adhere to either of these trends. The relatively high degree of variation in genomic repeat content across remarkably short evolutionary time scales in squamates presents the greatest contrast with birds and mammals. Unlike the clade-specific pattern observed in mammals, the genomic repeat content variation of squamate reptiles exhibits a high degree of variation even between species within the same genus (e.g., within the genera *Ophisaurus* (44.8–48.9%), *Coniophanes* (59.4–73%), and *Crotalus* (35.3–47.3%); Fig. 1c, Supplementary Figs. 2 and 3, and Supplementary Data 4). Across the 66 squamate species sampled, total genomic repeat element content varied from 24.4% to 73.0% (3-fold variation; Fig. 1c). Collectively, our analyses highlight the remarkable finding that the comparatively small genomes of squamates, similar to those of birds, can contain large and highly variable amounts of repeat elements, exceeding the range reported for mammals.

### Genomic TE composition across squamate reptiles.
The content and evolutionary dynamics of TEs in squamate genomes are unique in many ways when compared to that of mammals and birds, yet squamate genomes also share several key features with both lineages. All three groups have TE landscapes largely dominated by non-long-terminal repeat (non-LTR) retrotransposons. However, unlike mammalian genomes in which L1 LINEs and associated short interspersed nuclear elements (SINEs) are the most dominant and active elements[3,15], squamate genomes tend to contain three similarly abundant and active LINE families (CR1, BovB, and L2 LINEs; Fig. 1, Supplementary Fig. 2, and Supplementary Data 4). While CR1 LINEs are ubiquitous across amniote genomes, CR1s are particularly abundant and recently active in squamate genomes (5.1%, compared to

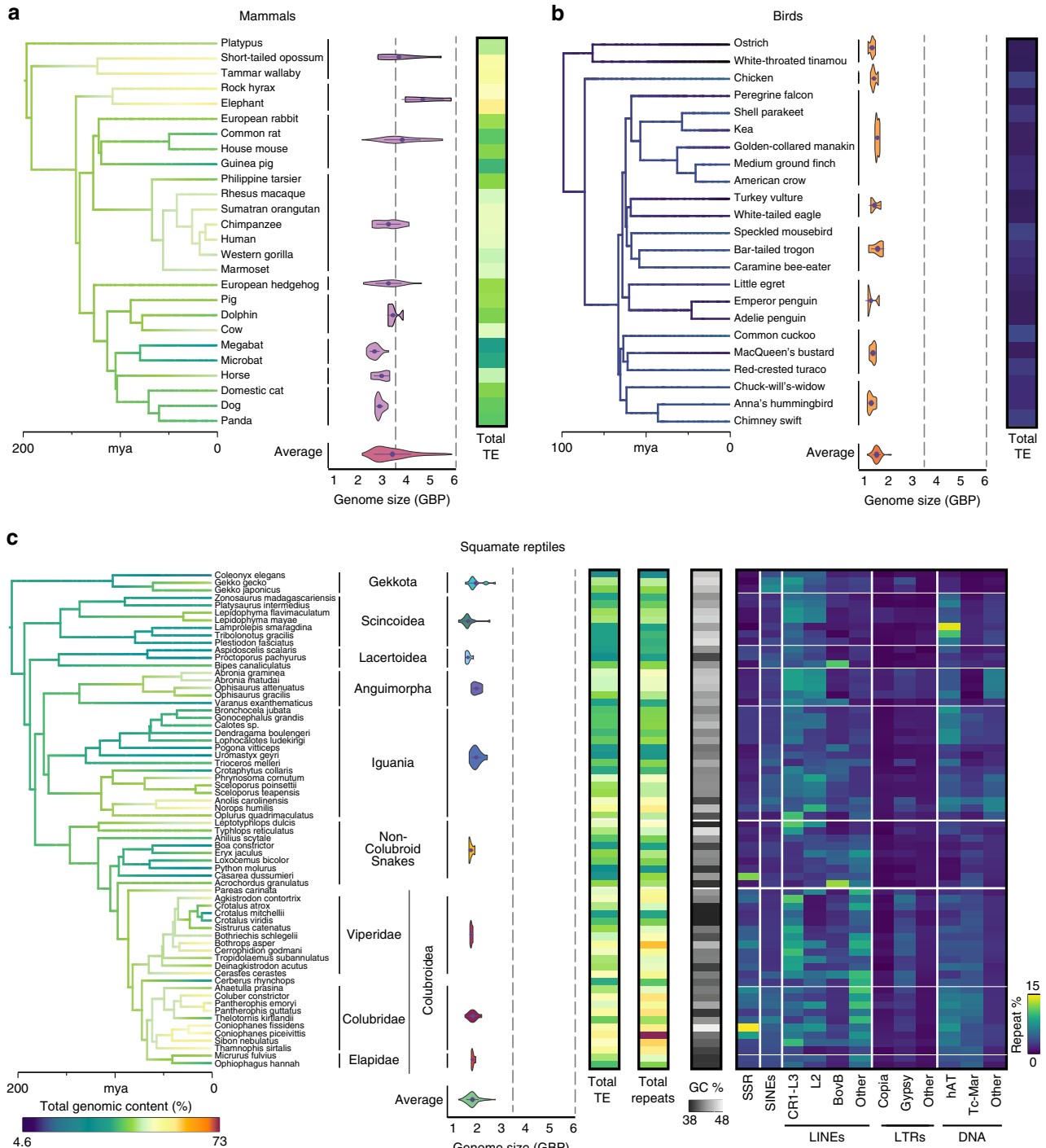

**Fig. 1** Genomic transposable element (TE) abundance and genome size variation in mammals, birds, and squamate reptiles. Branches on the time-calibrated consensus phylogeny are colored according to the estimated rate of genomic TE evolution. Violin plots show distributions of flow cytometry-based genome size estimates for major groups of **a** mammals, **b** birds, and **c** squamate reptiles, and the associated heat maps reflect the total genomic TE content (%) for each taxon. For squamate reptiles, additional heat maps show percent genomic repeat element content, percent genomic GC content, and percentages of major components contributing to the overall repeat element landscape

~3.5% in birds and <1% in mammals[1]), as they tend to be in other non-avian reptiles (i.e., ~10% in crocodilians[16]). In addition to non-LTR elements, DNA elements are also highly variable and particularly abundant in multiple divergent squamate lineages (Fig. 1). For example, Tc1-Mariner elements have experienced a 2.4-fold expansion in colubroid snakes compared to lizards (mean genomic abundance = 4.23% in colubroid snakes and 1.7% in lizards; Fig. 1, Supplementary Fig. 2, and Supplementary Data 4).

The most striking contrast between squamate vs. bird and mammal genomes is that squamate genomes contain an unusually broad diversity of types, subtypes, and families of TEs that appear simultaneously active[12,16–19] (see also below and Supplementary Fig. 3), whereas genomes of mammals and birds tend to have a very small number of active elements (e.g., L1 LINEs and Alu SINEs in mammals, and endogenous retroviruses (ERVs) in birds[6,15,20,21]).

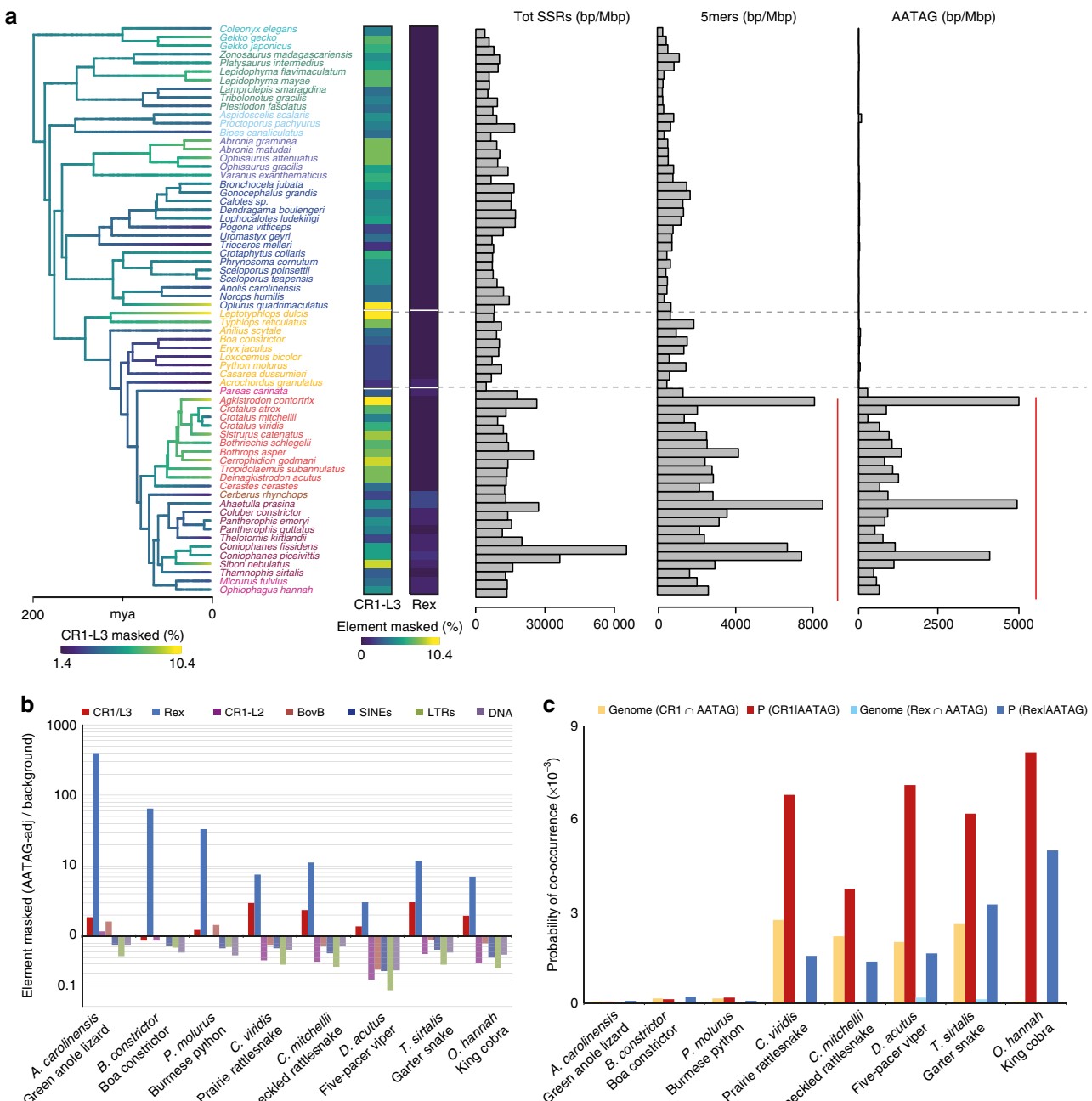

**Fig. 2** Microsatellite seeding by transposable elements (TEs) in squamate reptiles. **a** Branches on the time-calibrated consensus phylogeny are colored according to estimated rates of genomic CR1-L3 LINE evolution. Heat maps show the total genomic content (%) of LINE retrotransposon types involved in microsatellite seeding. Associated bar plots represent the total (left), 5mer (middle), and AATAG (right) microsatellite bp/Mbp density frequencies for each genome sampled. Red lines to the right of the bar plots highlight pronounced seeding of 5mer and AATAG microsatellites in colubroid snakes. **b** The ratio between TE mapping at the 5′ tail of AATAG microsatellite loci (AATAG-adjacent) and TE content averaged over five independent, randomly simulated genomic backgrounds for each class of TEs (SINEs; CR1-L3, Rex, CR1-L2 and BovB LINEs; LTRs; and DNA transposons). Ratios are plotted on a log scale to highlight enriched elements flanking AATAG loci (ratio >1) in contrast to elements more abundant in the genomic background (ratio <1). **c** Histogram shows joint and conditional probabilities of associations between AATAG loci and CR1-L3 and Rex. Genomic joint probabilities are shown in orange and light blue for CR1-L3 and Rex, respectively. AATAG-adjacent conditional probabilities are shown in red and dark blue for CR1-L3 and Rex, respectively

Guanine-cytosine (GC) content is known to play an important role in genome and repeat element evolution[22–26]. We found evidence of significant relationships between GC content and total TE content, as well as GC and total microsatellite (or simple sequence repeat; SSR) content, in lizards and colubroid snakes (Supplementary Fig. 4). In contrast, we found no correlation between genomic GC content and any aspect of the genomic repeat element landscape in non-colubroid snake genomes (Supplementary Fig. 4). Consistent with previous studies[13], our analyses highlight the surprisingly variable nature of GC content across squamate genomes, which tends to be higher in lizards than in snakes, yet highest in the colubroid snake *Coniophanes fissidens* (GC = 47.8%; Fig. 1c). These findings are also broadly consistent with previously reported shifts in GC isochore

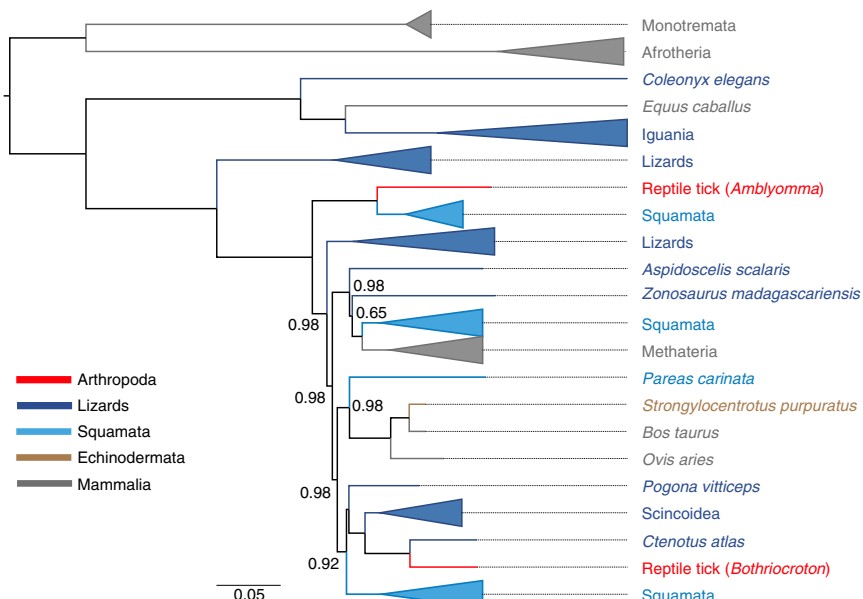

**Fig. 3** Evidence for ectoparasite-mediated horizontal transfer of BovB LINEs in squamate reptile genomes. A summarized Bayesian phylogenetic tree of full-length BovB LINE sequences for 87 metazoan species, including two reptile ticks. Branches have been collapsed and colored to represent major clades. Posterior probabilities are shown only at nodes that had posterior support <0.99

structure in squamate genomes[17,26], including the absence of isochore structure in lizard species, and intermediate structure in snakes that appears to represent isochore reacquisition after isochore loss in a squamate ancestor[13].

**Unparalleled microsatellite abundance in squamate genomes.** Our analyses revealed that some squamate genomes contain astonishingly high levels of SSRs, and that genomic SSR content in some snake species is the highest of any previously studied vertebrate (e.g., 14% according to RepeatMasker estimates in *Coniophanes fissidens*, Supplementary Data 4 and 5, and Supplementary Fig. 5). While previous studies have suggested that the highest variation in SSR content tends to exist among major vertebrate lineages[27], with fish, squamate reptiles, and mammalian genomes having similarly high genomic content[12,13,17,28], our results provide new evidence that the highest variation known in genomic SSR content exists within lineages—squamates and snakes, specifically. We found up to 10.9-fold variation in the genomic density of SSR loci (262–2845 loci/Mbp) and 16.6-fold variation in SSR-occupied bases per Mbp (4.08–67.94 Kbp/Mbp) among squamates overall, with non-colubroid snakes tending to have the lowest genomic SSR abundance, and colubroid snakes having the highest (Supplementary Data 5, Fig. 2, and Supplementary Figs. 5 and 6). This extreme variation in the genomic SSR content of squamate reptiles exceeds the previous high benchmark set by fish genomes (8.2-fold loci/Mbp and 18.0-fold bp/Mbp variation), and dwarfs that of mammals (5.8-fold loci/Mbp and 5.4 bp/Mbp) and bird genomes (1.8-fold loci/Mbp and 2.8 bp/Mbp)[12,13,17,28].

**Largest instance of microsatellite seeding among vertebrates.** A peculiar feature of SSR evolutionary dynamics in squamate genomes is the significant shifts in 4mer and 5mer abundances across the squamate tree, including extreme expansion of specific 4mer and 5mer SSRs motifs in colubroid snake genomes (Kruskal–Wallis test $p$ value <0.001, Supplementary Fig. 6 and Supplementary Data 6). Two specific SSR sequence motifs, ATAG and AATAG, account for most of the microsatellite expansion in colubroid snakes, representing a 7.4-fold increase in ATAG (bp/Mbp) and an 87.7-fold increase in AATAG (bp/Mbp) compared to the averages of other squamate genomes (Supplementary Figs. 7 and 8). The extremely high genomic representation of these two similar SSR sequence motifs in snake genomes suggests a motif-specific mechanism has driven their expansion. Previous studies[12,13] have suggested that LINE retrotransposons that contain microsatellites on their 3′ end in snakes might lead to SSR genomic expansion through a process called "microsatellite seeding."

To test the hypothesis that microsatellite seeding is responsible for the expansion of particular SSR sequence motifs, we surveyed the regions adjacent to the two most highly expanded SSR motifs (AATAG and ATAG) in eight complete reptile genome assemblies. Consistent with the expectations of microsatellite seeding, we found strong statistical support that CR1-L3 LINEs tend to be immediately adjacent to AATAG loci in colubroid genomes (Fisher's exact test $p$ value $<2.2e^{-16}$), as well as strong statistical enrichment of AATAG loci at the 3′ end tail of Rex LINEs ($p$ value $<2.2e^{-16}$) in all squamate genomes sampled, suggesting that both CR1/CR1-L3 and Rex LINEs contribute to microsatellite seeding in squamate genomes (Fig. 2b and Supplementary Data 7). In contrast to elements adjacent to AATAG repeats, we found no evidence of enrichment in adjacency for any particular TE for the second most expanded SSR motif (ATAG) compared to randomly sampled genomic regions; this suggests that the expansion of this motif is not directly driven by microsatellite seeding, although its similarity to AATAG suggests that it might be indirectly related. To further identify the specific LINE element that is responsible for microsatellite seeding of AATAG SSR loci, we calculated the conditional probability of TE-SSR co-occurrence in a genome-wide context compared to the AATAG-adjacent context. Conditional probabilities of AATAG loci and CR1-like LINEs genomic co-occurrence are noticeably different only for CR1-L3 LINEs between colubroid snakes and other squamates (Fig. 2c), and are only barely detectable for Rex LINEs. Additionally, CR1 LINEs are a major contributor to the genomic TE landscape of squamates (particularly colubroid snakes), whereas Rex elements

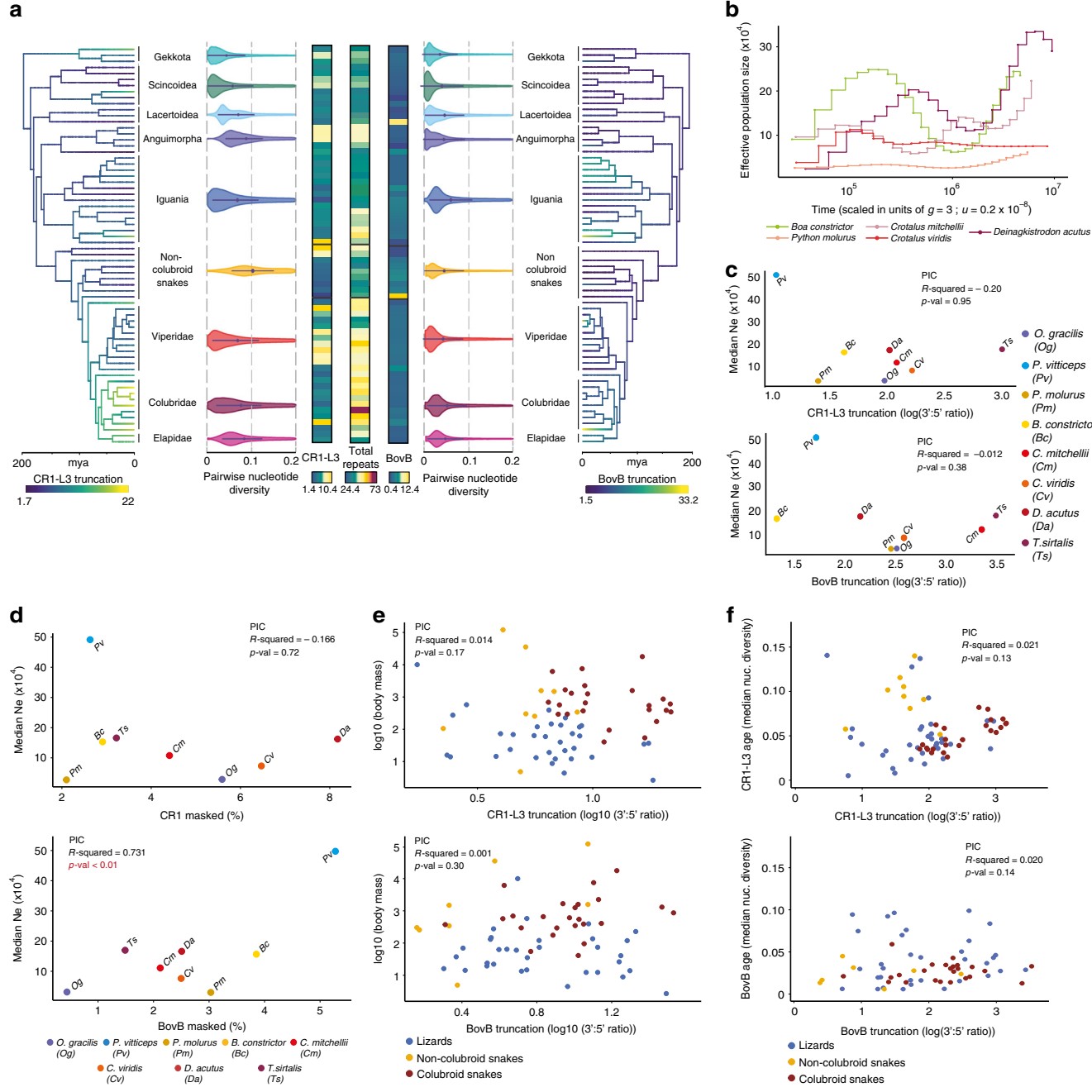

**Fig. 4** Relationships between truncation, effective population size, body mass, and divergence estimates for CR1-L3 and BovB LINE retrotransposons among squamates. **a** Branches on the time-calibrated consensus phylogenies are colored according to the calculated 3′:5′ read depth coverage ratio for CR1-L3 (left) and for BovB (right) LINEs. Heat maps show the genomic content of CR1-L3 LINEs, total repeats, and BovB LINE retrotransposons represented as percentages of the total genome. For each major clade, violin plots show the density distributions of divergence estimates (pairwise $\pi$) for all CR1-L3 and BovB elements compared to the species-specific consensus sequence. **b** Variation in effective population size ($N_e$) over time for five snake species scaled by generation time and mutation rate ("$g$" and "$u$" on the x-axis). **c** Relationship between $N_e$ and truncation of CR1-L3 (top) and of BovB (bottom) LINEs for eight squamate species. **d** Relationship between total genomic abundance of CR1-L3 (top) and BovB (bottom) LINEs and $N_e$. **e** Relationship between adult body mass and degree of truncation across 66 squamate species for CR1-L3 (top) and BovB (bottom) LINEs. **f** Relationship between age (median pairwise $\pi$) and truncation for CR1-L3 (top) and BovB (bottom). Summary statistics from phylogenetically independent contrasts (PIC) are shown as insets for each plot in **c**–**f**. Statistical analyses were performed after log transformation of truncation values in plots **e** and **f**

represent a very small fraction. Taken together, our data indicate that microsatellite seeding may be a common ancestral feature of multiple families of squamate LINEs, yet the high activity and expansion of CR1-L3 LINEs has driven associated AATAG loci to extremely high frequencies in colubroid snakes, leading to an astounding 74.73-fold genomic AATAG loci/Mbp increase in this lineage, and the highest levels of genomic SSR content among

vertebrates. The ramifications of such extreme levels of homologous SSRs in colubroid snakes, in terms of genome function and evolution, remains uninvestigated. A potential role in mediating increased ectopic recombination leading to gene duplication has been suggested by previous studies that have identified an enrichment of these repeats surrounding tandemly duplicated venom genes in snakes[12,29,30]. Collectively, these findings imply

the exciting possibility that LINE-SSR hybrid elements may have played key roles in the evolution of prominent phenotypes in snakes (i.e., venom evolution).

**Multiple independent TE horizontal transfer events.** Evidence for the horizontal transfer of BovB LINEs has been identified by previous studies[12,31–34], and our analysis of squamate genomes provides new insight into the complexities of BovB horizontal transfer. Our phylogenetic reconstruction of BovB LINEs, including samples from our squamate genomes and other sequences from GenBank[35], highlights multiple horizontal transfer events, and supports ectoparasite-mediated transfers of BovB LINEs into and out of squamate reptile genomes (Fig. 3 and Supplementary Fig. 9a, Supplementary Data 9). We found BovB LINE sequences from squamate species clustering with other groups of metazoans in all branches of our phylogenetic tree, consistent with multiple horizontal transfer events of BovB from lizards to mammals and to other squamates, and from snakes to mammals and other squamates. Previous studies found support for virus-mediated transfer of TEs[36], and suggested ectoparasites as potential transmission vectors[34,37–40]. Our analyses support the horizontal transfer of BovB from one reptile tick species (*Amblyomma limbatum*) to colubroid snakes (Supplementary Fig. 9a), and provide the first ever evidence for ectoparasite-mediated transfer from squamate genomes in the case of the reptile tick *Bothriocroton hydrosauri*. Samples containing BovB elements sequenced from this tick species are deeply nested among lizard-derived BovB sequences, yet are unique in containing a large internal deletion (1691 nt) relative to all other lizard-derived BovB sequences in this clade. Collectively, our analyses of BovB LINE evolution showcase a dynamic history of horizontal transfer that encompasses essentially all forms of the process of transfer into and out of squamate genomes, implicating the role of ectoparasites in both directions of the transfer process.

**Testing explanations of variation in genomic TE abundance.** Multiple studies have suggested that purifying selection acting against TE insertions may manifest in correlations between $N_e$ and features of the genomic TE landscape. This prevailing demographic explanation for variation in repeat content has been invoked to describe patterns of genome complexity and evolution across the tree of life, and predicts that lineages with higher $N_e$ should undergo more effective purifying selection and thus lower genomic accumulation of mutationally hazardous DNA[41,42]. Indeed, previous population (within-species) and phylogenetic (among-species) studies have provided rationale and empirical evidence that TE insertion rates, fixation rates, and abundance may be correlated with $N_e$[14,42–45]. Relative insert length has also been linked to population size at the population level by an ectopic recombination model in which element length is correlated with the strength of selection[14,18,43,46–48].

Using our phylogenetic-scale dataset, we tested if features of TE landscapes (i.e., genomic abundance, estimated age of activity, and degree of truncation for BovB and CR1-L3 LINEs) showed evidence of a correlation with estimates of $N_e$ consistent with a demographic model of TE landscape evolution. We first tested for a relationship between $N_e$ and TE landscape characteristics using the median values of $N_e$ estimates derived from pairwise sequentially Markovian coalescent (PSMC) analyses[49] for eight published squamate genomes (Fig. 4b–d and Supplementary Fig. 10). With this dataset, we found no evidence supporting a correlation between $N_e$ and CR1-L3 and BovB length or genomic repeat element abundance (Fig. 4c, d and Supplementary Fig. 10c–e). Notably, we found that species with similar $N_e$ estimates (Fig. 4b) showed different levels of truncation and of TE

genomic abundance, and that even within a species TE truncation and abundance were poorly correlated (Fig. 4a, c, d and Supplementary Figs. 10 and 11). Second, to further test for correlations between $N_e$ and element abundance or truncation using an approach that is independent of inferences of generation time and mutation rates, and independent of potential biases associated with coalescence-based estimates of $N_e$ (i.e., population substructure, migration, selection)[49–55], we used adult body mass as a proxy for $N_e$ for all species included in our study (as in ref. 56; Supplementary Data 8)[57]. This approach has the added benefit of leveraging the much larger sample size of our entire dataset (compared to our PSMC analyses using eight complete genomes). Similar to our PSMC-based analyses, we compared body mass to CR1-L3 and BovB genomic abundance, their degree of truncation, and total genomic repeat element and TE abundances. Consistent with our PSMC-based analyses, we failed to find a correlation between body mass and truncation (Fig. 4e and Supplementary Fig. 12b) that would support a demographic model of TE landscape evolution; the only correlative trend that we did find was a correlative trend that is opposite of that predicted by the demographic model between $N_e$ and genomic repeat element abundance instead (i.e., higher $N_e$ was positively correlated with TE abundance; Supplementary Fig. 12d). Finally, to test more generally for evidence that selection acts on TE length at the phylogenetic scale, we tested for a link between TE truncation and TE age[18,48,58] using median pairwise divergence of TE copies from their subfamily consensus, $\pi$, as a proxy for age for CR1-L3 and BovB families, and found no correlation (Fig. 4f, Supplementary Fig. 13, and detailed in Supplementary Figs. 14–16). While we acknowledge the complexity of testing links between two highly dynamic evolutionary processes (e.g., $N_e$ and TE abundance), and the limitations of methods used to make inferences about these processes (i.e., $N_e$ estimation), all of our analyses fail to provide support for $N_e$ as a strong determinant of variation in the composition and characteristics of the repeat element landscape at the phylogenetic level across squamate reptiles. Although our analyses cannot fully reject a demographic hypothesis that a relationship between $N_e$ and TE characteristics exists (i.e., we can only fail to reject a lack of relationship), the apparently poor explanatory power of the demographic hypothesis in predicting squamate TE activity and abundance suggests that perhaps other factors, such as variation in molecular mechanisms of TE proliferation, silencing, and removal, may better explain the majority of variation in TE abundance at the phylogenetic level in squamates.

## Discussion

This broad glimpse into the diversity of repeat structure and composition of squamate reptile genomes suggests that this lineage possesses particularly distinct and often extreme repeat landscape characteristics compared to other amniotes. Our results provide evidence for surprisingly high variation in the content and composition of genomic repeat elements across squamate lineages, including 3-fold variation in the identifiable genomic repeat element content. We also discovered that some snake genomes have experienced microsatellite expansion at unprecedented scales through the process of microsatellite seeding by specific LINEs, leading to genomic microsatellite abundances that are the highest of any known vertebrate genome. Despite such extreme variation in genomic repeat element content, genome size across squamates is remarkably conserved (~0.2-fold variation), challenging the prevailing view that genomic repeat abundance and genome size tend to tightly co-evolve[4]. These findings provide some of the strongest evidence for a dynamic equilibrium or an "accordion" model, in which genomic DNA gain through

TE expansion may be approximately balanced by genomic DNA loss through deletion[7,58,59]. Overall, these results highlight extreme shifts in the structure of squamate reptile genomes, and further beg the question of whether particular aspects of squamate genome function and evolution are also more unique and variable compared to other vertebrates. These findings argue that squamates may represent a particularly powerful model system for testing hypotheses about genome structure, function, and evolution, and their interactions.

Many previous studies focused on population-level dynamics of TE evolution have shown that differences in $N_e$ and the efficacy of purifying selection acting against TE proliferation have played a major role in structuring the repeat landscape of many eukaryote genomes[9,18,46–48,58,60–63]. Even in squamate species (e.g., *Anolis* lizards), variation in $N_e$s has been linked to TE insertion length and fixation probability[18,62,63]. Our phylogenetic-scale analyses across squamate species, however, recovered no clear evidence linking genomic repeat abundance or activity with $N_e$ estimates in squamates. Although coalescent-based estimates of $N_e$ can be biased by a number of model violations (i.e., population substructure, selection), we also failed to find a significant relationship between genomic repeat characteristics and body mass—a known correlate of $N_e$. Population size is, however, likely to have influenced other aspects of genome evolution, such as fixation of deletions, that could contribute to the maintenance of nearly constant genome size in squamates.

Our results together with those from previous studies suggest that different evolutionary forces may dominate different evolutionary scales, and that while demographic processes (and purifying selection) may dominate population-level trends in TE evolution, phylogenetic-scale patterns in TE landscapes may be more strongly determined by other processes. Evidence for extreme variation in transcriptional levels of TE-derived transcripts across squamates[12], together with evidence from this study of lineage-specific swings in repeat element proliferation, suggest that molecular mechanisms related to TE regulation may be particularly relevant at the phylogenetic scale in squamates. Squamates may, therefore, represent a valuable system for studying the impacts of variation in molecular mechanisms of TE control, such PIWI-interacting RNA dynamics and efficacy, epigenetic silencing of TEs, lineage-specific TE activity, DNA repair mechanisms, and post-insertion 5′ removal of TEs. Further studies are needed to address the question of whether variation in molecular mechanisms of TE silencing and activity, as well as DNA repair, explain variation in squamate genomic TE content, and would provide fascinating insight into the factors that shape genomic repeat landscape variation.

## Methods

**Taxon sampling and library preparation**. DNA extraction of 52 squamate samples (total = 45 species) was performed using a phenol–chloroform–isoamyl alcohol (PCI) extraction protocol. Random shotgun genome libraries were prepared by fragmenting DNA samples to an average length of 300–600 bp using a M220 Covaris Ultrasonicator. The NEBNext Illumina DNA Library Prep Kit (New England Biolabs) was used following the manufacturer's protocol to perform fragment-end repair, poly-A tailing, adapter ligation, and library amplification. After library preparation, fragments were size-selected using a BluePippin (Sage Science) for a length of 350–450 bp. Pooled multiplexed libraries were sequenced on an Illumina MiSeq with 300 bp paired-end reads. Paired reads were merged based on sequence overlap and were adapter and quality trimmed using CLC genomics workbench 9.0.1[64]. Roche 454 shotgun sequencing data of nine snake species from previous studies[12,13] and draft genome assemblies of 12 additional squamate species (Supplementary Data 1) were also included. Our final sampling included a total of 66 different squamate species. For each species, mitochondrial reads were filtered out in CLC genomics workbench 9.0.1 using the complete mitochondrial genome of the most closely related species available on GenBank[35]. Reads that mapped to the reference were used to assemble species-specific mitochondrial genomes. Reads that did not map to the reference (i.e., nuclear reads) were used for downstream repeat element annotation and analyses.

**SSR identification and analysis**. We used Pal_finder v.0.02.03[65] (Palfinder hereafter) to identify microsatellites. Default Parfinder parameters were used to identify perfect dinucleotide (2mer), trinucleotide (3mer), and tetranucleotide (4mer) that were tandemly repeated for a total length of at least 12 bp. Perfect pentanucleotide (5mer) and hexanucleotide (6mer) tandemly repeated motifs were annotated only if longer than 15 bp. Loci/Mbp and bp/Mbp frequencies were calculated for all microsatellite motifs, length classes (2–6mers), and total content, and summarized per genome and major taxonomic group. Tests for multiple evolutionary rates of microsatellite abundance across lineages, ancestral state reconstruction of genomic microsatellite frequencies, and quantification of microsatellite landscape differentiation among species were performed using the R packages Phytools v.0.4–60[66] and APE v.3.3[67]. For the multiple evolutionary rate analysis of microsatellite (and TE) abundance, we conducted censored rate tests using Phytools with 1000 simulations (to compute $p$ values) on 100 randomly sampled posterior trees using the restricted maximum likelihood technique to obtain unbiased estimates of the evolutionary rate parameter ($\sigma$)[28]. We used the time-calibrated phylogeny and the *pic* function in R (provided by the APE package) to compute phylogenetic independent contrasts for tests of clade-specific differences in genomic microsatellite content. We performed the nonparametric Kruskal–Wallis $H$ test in R after the data rejected normality (Shapiro–Wilks test; $p$ values <0.05 before and after log transformation) and homogeneity of variances (Bartlett's test; $p$ values <0.05 before and after log transformation). Between lineages variation was tested using a post hoc Dunn test for multiple comparisons using the Benjamini–Hochberg correction method in R (Supplementary Data 6).

**TE identification and analysis**. Squamate genomic repeat elements were annotated according to homology-based and de novo identification approaches. Because repeat element annotation can be highly dependent on the repeat library used, we built large multi-species (clade-specific) repeat libraries that we used to annotate repeats for all members of a clade. To build these clade-specific libraries, we first performed de novo repeat element annotation on each species (except where already published) using RepeatModeler v.1.0.9[68], followed by further repeat classification in CENSOR[69]. Second, we built clade-specific de novo repeat element libraries, one for all lizard species (33 species de novo reference library) and one for all snake species (de novo TE libraries for 21 species were combined, and merged with the reference library generated by Castoe et al.[13]). Each clade-specific library was then filtered to avoid redundancy of highly similar elements. We tested whether using a single squamate-specific library for all species would change the inferred relative TE content and overall amount of repeat identified; we found no detectable difference between the results of the two masking protocols (Supplementary Fig. 17), and therefore decided to use the two clade-specific libraries in order to reduce masking time by reducing the overall library size. Additional classification of unknown (unclassified) elements was achieved by comparing these unclassified elements to all elements that were classified using BLAST[70]. Additionally, we generated squamate-specific BovB and CR1-L3 LINEs reference sequence libraries for all 66 species included (additional information regarding library generation are provided in the following paragraph).

Repeat element analyses were performed in RepeatMasker v.4.0.6[71] with default parameter settings. To maximize element identification, we used a custom bash script to specify the order of the four libraries used as references for the masking process: (i) BovB-L3 LINEs library, (ii) Tetrapoda *RepBase* library (version 20.11, 07 August 2015[72]), (iii) classified elements from the clade-specific library for either snakes or lizards, and (iv) unknown elements from the clade-specific library. We used the BovB-L3 LINEs library first to control for limited sampling and low-quality reference sequences of squamate reptile BovB and L3 LINEs in the tetrapoda library. RepeatMasker output files were post-processed using a custom-modified implementation of the *ProcessRepeat* script included in the RepeatMasker package. Specifically, we modified the output to include additional summary information in the *.tab* output file for TE subfamilies that are important and/or frequent in squamate reptiles (e.g., CR1-L3, L2, and Rex). Also, because the provided *ProcessRepeat* script still reflects old and outdated classification schemes of TEs (e.g., Penelope elements are inappropriately classified as LINEs), we made other modifications to the *ProcessRepeat* script to correct for such errors according to the classification reported by Chalopin et al[6].

**Comparing sampled and assembled genomes**. We tested whether genomic repeat content estimated from unassembled shotgun genomic datasets were similar to estimates derived from fully assembled genomes. We compared RepeatMasker estimates of total TE genomic abundance between assembled genomes and unassembled shotgun genomic datasets for the same species (*Python molurus*, *Boa constrictor*, *Thamnophis sirtalis*, and *Deinagkistrodon acutus*) or for two closely related species belonging to the same genus (*Gekko gecko* vs. *Gekko japonicus* and *Ophisaurus attenuatus* vs. *Ophisaurus gracilis*). We also tested for potential biases due to unequal genomic sampling in the shotgun datasets. We extracted at random subsamples of 3, 5, 8, 10, 30, 50, 100, and 250 Mbp from unassembled genomic shotgun datasets of four species (*Python molurus*, *Gekko gecko*, *Ophisaurus attenuatus*, and *Pantherophis emoryi*), and compared RepeatMasker estimates of total TE genomic abundance for each. Read extraction was performed using the *subsample_fasta.py* script from the QIIME pipeline[73]. Finally, we compared RepeatMasker estimates of total TE genomic abundance in relation to the amount

of sequence data obtained for all Illumina and 454 genomic shotgun datasets to test for biases related to sequencing technology, and for biases related to the amount of sequence data collected per individual, vs. estimates of total TE genomic abundance.

**CR1 and BovB LINEs phylogenetic and evolutionary analyses.** Species-specific consensus sequences for both CR1-L3 and BovB LINE retrotransposons were generated in CLC genomic workbench 9.0.1 using default parameters, a linear gap cost, and the global alignment setting. Nuclear reads for each species were mapped to the consensus sequence of the LINE consensus sequence from the most closely related species available, which was used as initial reference (e.g., both CR1-L3 and BovB reference sequences for the Burmese python were generated by Castoe et al.[13] and used as reference for building the consensus for the Mexican burrowing python). The first consensus generated was then used as a new reference for further rounds of re-mapping of nuclear reads until no additional mapping reads were recovered. Consensus sequences were determined by simple majority rule consensus, removing regions with coverage <10x after the second mapping iteration, and <20x in the final mapping. Consensus sequences were aligned in ClustalW[74] with a gap open penalty of 50, and alignments were manually adjusted prior to downstream analyses (Supplementary Data 10). To the CR1 consensus sequences generated from our 66 squamate species, we added CR1-L3 and CR1-L2 vertebrate consensus sequences available in *RepBase*, for a total of 155 sequences (Supplementary Data 10). Squamate BovB consensus sequences we generated from our 66 squamates were combined with other metazoan consensus sequences available in *RepBase*, for a total of 87 sequences (Supplementary Data 9). Bayesian phylogenetic tree reconstruction analyses of squamate CR1 and BovB LINEs were performed in BEAST2[75]. Two independent analyses were run for 200 million generations each, following the Yule model of speciation and a relaxed log-normal clock model; MCMC chains were sampled every 1000 generations. The program Tracer v.1.6[76] was used to confirm that the MCMC chains had reached convergence. We conservatively discarded the first 25% of collected MCMC generations as burn-in, based on evidence that the likelihood and parameter values reached stationarity after approximately 15% of the sampling process.

**CR1 and BovB LINEs coverage and age analyses.** For each species, the species-specific CR1-L3 and BovB consensus sequence was used as a reference to estimate read coverage using the BWA *mem* alignment tool[77], and the BEDTools2 (version 2.26.0) *coverage* tool[78]. Coverage counts were normalized by the total number of reads aligned to the full-length reference sequence. Read coverage was estimated for: (i) each 10 bp sliding window, (ii) for the first and second half of the reference sequence, and (iii) for each third of the reference.

We used pairwise sequence divergence from the consensus (pairwise $\pi$) as a proxy to infer age and relative element level of activity through time. Pairwise distances values for each element and species were estimated following a custom pipeline starting from BWA alignments. An R[79] custom script built on the pegas[80] and stringr packages was used to calculate pairwise $\pi$ estimates using multi-fasta pairwise alignments of reads to the reference. Because we expected multiple TE subfamilies to exist, sequence divergence was estimated by excluding sites that define different CR1 and BovB subfamilies. For each species, we calculated the relative nucleotide frequency for each position in the multiple sequence alignment, and then calculated the mode of the frequency distribution (bins of 0.01) of the most frequent nucleotide at each position. Sites for which the most frequent nucleotide was in a bin more than three bins away from the mode were discarded as defining a separate subfamily.

**Time-calibrated phylogeny of 66 squamate reptiles.** We estimated a time-calibrated phylogeny for the 66 squamate species in our study and a additional eight outgroup vertebrates. We downloaded and parsed 12 mitochondrial-encoded protein-coding genes for each species with a mitochondrial genome sequence available on GenBank. The same genes were parsed from our de novo assembled mitochondrial genomes after genes were annotated for these using MITOS[81]. We aligned the 12 protein coding genes encoded on the mitochondrial heavy strand using MUSCLE v.3.8.21[82] and concatenated the sequences into an alignment that we used for divergence dating (10,479 bp). Prior to divergence dating, we estimated the best-fit partitioning scheme and associated models of nucleotide substitution using Bayesian information criterion and the heuristic search algorithm provided in PartitionFinder v.1.1.1[83]. We provided a starting partitioning scheme that defined 36 partitions (splitting codon positions for each of the 12 genes), and PartitionFinder identified the best-fit partitioning scheme comprising a single partition for each codon position (three total) and a GTR+I+G model for each partition. We estimated divergence times using BEAST v.2.3.4[84] with a calibrated Yule model of speciation and a log-normal relaxed clock model. We constrained the topology to that provided from previous studies of the squamate phylogeny and diversification[85,86]; we also constrained divergence times of a total of seven nodes using fossil calibrations also provided in previous studies. Calibration points and associated prior distributions are given in Supplementary Table 1. Two independent MCMC runs were conducted for 100 million generations each, with MCMC chain sampling every 10,000 generations. We assessed convergence to the posterior based on likelihood and parameter stationarity (effective sample size >200 for all

parameters) using the program Tracer. We discarded the first 10% of generations as burn-in, based on the likelihood and parameter values exhibiting stationarity before 10% of sampling was completed.

**AATAG microsatellite seeding by TE analyses.** We performed adjacency analyses of AATAG and ATAG SSR loci on high-quality assembled genomes for seven snake species, and used the green anole lizard as an outgroup. To increase specificity, genomes were first masked only for simple repeats. We extracted coordinates of annotated AATAG and ATAG SSR loci from the *.out* RepeatMasker output files, and used these coordinates to extract target regions 400 bp upstream and downstream of each microsatellite locus. We then performed a second run of RepeatMasker to mask only TEs located in the extracted target regions that flank AATAG and ATAG loci. Following this strategy, we were able to annotate TEs located in close proximity to SSR loci, and to differentiate TEs that harbor microsatellite-like regions in their reference sequences. The composition of TEs physically associated with SSR loci regions was then compared to the average of five independent randomly generated genomic backgrounds matching in sample size the corresponding microsatellite landscape. For each species, genomic background reads were generated by using the *random* tool in the BEDTools2 v.2.26.0 package, in which we specified the number of sequences to be extracted and that their length was to match the SSR-adjacent genomic subsample. The generation of random bed files was performed independently five times per species, the TE composition was averaged across these five genomic backgrounds, and then compared to SSR loci adjacent regions. Fisher's one-tailed exact tests were performed to evaluate the enrichment of TE families in SSR loci regions (at $\alpha = 0.01$). Finally, to identify the specific element types involved in microsatellite seeding, we estimated genomic and SSR-adjacent conditional probabilities of TE-SSR co-occurrences. We estimated the conditional probability of sampling an AATAG SSR with an adjacent CR1 LINE present within 400 bp, and compared this to the estimated joint probability of sampling an AATAG SSR locus and a CR1 LINE using the genome-wide frequencies. We also calculated the conditional and joint probabilities for Rex LINEs, and compared those to the conditional and joint probabilities of CR1 LINEs, respectively.

**Effective population size ($N_e$) estimation.** Whole genomic Illumina paired-end reads for eight squamate reptiles species were first preprocessed for quality using Trimmomatic[87]. Clean paired and unpaired reads were aligned to their respective reference genome assemblies using BWA v.0.7.12, and single nucleotide polymorphisms were called with SAMtools (v.0.1.18) *mpileup*[88]. We applied the PSMC[49] using a generation time of 3 years across all eight species (which represents the average of generation time approximations available from the literature; Supplementary Table 2) after verifying that the application of a single generation time yielded results consistent with estimates of average $N_e$ produced by the application of generation times within the range reported in the literature. Multiple studies have provided evidence of relatively similar mutation rates across lineages of squamates[13,89]. Therefore, in our PSMC analyses we used the generalized squamate mutation rate reported in Green et al.[89] of $2.4 \times 10^9$ /year/site (as estimated from 4-fold degenerate sites between anole and python). To test the robustness of inferred population size estimates, we conducted 100 bootstrap replicate analyses by splitting the scaffolds into smaller segments and randomly sampling the segments with replacement. Default outputs of the *psmc_plot.pl* script were used to graphically summarize $N_e$ changes over time estimations per each bootstrapped sample (Supplementary Fig. 10b).

Coalescent approaches for estimating $N_e$ and changes in $N_e$ over time (like PSMC) have several intrinsic limitations. Importantly, they rely on explicit assumptions of a single population coalescent model (without subdivision, gene flow, or selection) to estimate the time since the most recent common ancestor of alleles at each locus, as well as an assumed generation time and substitution rate. Population structure has been identified as one major factor that can bias PSMC-based estimates of $N_e$[50,52,90,91]. For example, the inferred trend in $N_e$ variation of a structured population can portrait either a bottleneck or an expansion in population size whether the alleles were sampled from the same subpopulation or from different subpopulations, respectively[51]. Episodes of natural selection can also bias estimates of $N_e$ obtained using PSMC, as selection can manipulate the rate of coalescence at specific loci that are directly or indirectly linked to targets of selection[54,55]. Given the nature of our data, we are not able to assess the presence and extent of population substructure or selection, and therefore cannot exclude that our PSMC estimates are immune to such biases. Additionally, PSMC has low power at recovering rapid changes in $N_e$, which may be incorrectly estimated to have occurred over a longer period of time, and cannot recover recent nor very ancient changes in $N_e$ (e.g., younger than ~10 kyBP and older than ~3 myBP for humans)[49,51]. Thus, we suggest caution when interpreting our PSMC estimates of $N_e$ and $N_e$ changes through time. However, we found low variance across bootstrapped $N_e$ estimates once the most recent and most ancient time points were removed, and patterns of expansion and contraction of $N_e$ are consistent with alternations of glacial and interglacial periods during the middle Miocene climate transition, the Pliocene and the Pleistocene[92]. In an attempt to reduce potential biases associated with PSMC estimates of recent and ancient changes in $N_e$, median $N_e$ values were calculated after removing the first and the last time points from each sample. We replicated

each analysis (see below) after applying different filtering schemes to the standard PSMC outputs (e.g., removal of 10 and 25% of time point data, and inclusion of only time points between 20 kyBP and 10 myBP). Since all tests provided the same conclusions, we report only analyses performed using median $N_e$ values that were calculated according to the original filtering scheme. Additionally, we replicated all of our analyses using adult body mass as a proxy for $N_e$[56] to avoid potential biases associated with our coalescence-based methods of $N_e$ estimation (i.e., Fig. 4e). For each of the 66 squamate species, we obtained adult body mass measurements from the literature[57] which were used to further test for a demographic explanation for variation in repeat content alongside coalescent-based estimates of $N_e$.

**Testing demographic explanations of repeat content variation.** We performed linear regression analyses to test for correlations between $N_e$ and LINE truncation, $N_e$ and genomic abundance of BovB and CR1-L3 LINEs, truncation and genomic abundance of repeats, and between truncation and estimates of ages of repeat element activity. We used the *pic* function in APE and the time-calibrated phylogeny to compute phylogenetic independent contrasts to be used for all linear regressions. These analyses were conducted for both the coalescent-based estimates of $N_e$ and adult body mass as a proxy for $N_e$. Since truncation values violated assumptions of normality and homogeneity of variance (Shapiro–Wilks test; $p$ values <0.05 and Bartlett's test; $p$ values <0.05), we performed statistical analyses on log-transformed values (Shapiro–Wilks test; $p$ values >0.05 and Bartlett's test; $p$ values >0.05).

**Data availability**. New raw, unassembled shotgun sequencing data and new assembled genome data have been deposited at NCBI under the following accessions:PRJNA413172 and PRJNA413201. The authors declare that all data and scripts used in this study are available via public databases or available from the corresponding author upon request.

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

## Acknowledgements

Support for this work was provided from startup funds from the University of Texas at Arlington to TAC. We acknowledge the Texas Advanced Computing Center (TACC) for providing access to computational resources.

## Author contributions

G.I.M.P. and T.A.C. designed and performed experiments, analyzed results, and wrote the manuscript; R.H.A., D.C.C., A.B.C., and D.R.S. collected data and analyzed results; J. R.-V. collected data; B.W.P., R.P.R., M.W.V., and J.A.S. analyzed results; all authors contributed to editing the final manuscript.

## Additional information

**Competing interests:** The authors declare no competing interests.

