## [Peer Review File · Nature Communications]

Reviewers' comments:

Reviewer #1 (Remarks to the Author):

The authors have amassed a rather impressive genomic dataset from more than 60 squamates. They demonstrate that lower-coverage datasets can accurately predict genomic repeat content. They show that some squamate genomes are comprised of more SSRs than any studied vertebrate, and that snake genomes have been seeded with microsatellites by LINEs to an unprecedented degree. They also use phylogenetic analyses to demonstrate multiple horizontal transfer events (most of which has already been documented), including a previously unidentified instance of snake-to-tick BovB transfer. They then test two competing (although not necessarily mutually exclusive) hypotheses about the forces controlling TE diversity and abundance in vertebrate genomes - one where species demography affects the strength of purifying selection against deleterious elements and another where the balance between DNA gain and loss mediate the amount of repetitive DNA that can accumulate in the genome. While the intended scope of the manuscript is notable, I believe the authors have missed some opportunities to provide strong evidence for some of their conclusions, in particular that current hypotheses explaining repeat content variation across vertebrates do not hold for squamates.

I am not wholly convinced that the lack of relationship between N_e and TE truncation as shown here rules out the role of demography in the accumulation success of TEs in a genome. For instance, BovB has been characterized in the *Anolis* genome, where it is very abundant and ancient (as calculated by average divergence from consensus) and likely extinct. Therefore, it is likely that these elements reached fixation in the *Anolis* genome long before the species evolved. Thus, they are probably more truncated because they are older and have been hit by more deletions over evolutionary time. You didn't include green anole in the demographic analysis here, but let's assume *Pogona* as an iguanian is similar. Your plot in Figure 4 and in the supplementary shows this - more truncation in BovB than CR1 in *Pogona*. In contrast, the subfamilies of CR1 are currently producing copies in the *Anolis* genome (i.e., low average divergence from consensus), these may potentially produce new full length copies and thus would be susceptible to purifying selection. However, I am not sure why you lumped together all CR1 subfamily elements within each genome to calculate divergence - it's mentioned in the supplement that you excluded subfamily-diagnostic sites but wouldn't there also be random mutations in each subfamily that occurred over evolutionary time? To me, this is potentially problematic, and it should be relatively simple to break CR1 up into families and calculate divergence within each. Thus, the estimates of age and activity for element families may be biased towards older ages and lower activity. Please either explain better how you did this or try to calculate divergence within subfamilies. I wonder if similar age classes of subfamily show the same amount of truncation.

Perhaps more importantly, the amount of 3':5' truncation wouldn't necessarily be an indication of the strength of purifying selection acting against elements - target primed reverse transcription is mistake-prone and more often than not leaves truncated copies, so we expect more truncation in any genome regardless of demographic history. Meanwhile,

purifying selection is a post-insertional force. Therefore, it would be more meaningful to look at the level of fixation in the genome - something entirely possible with your short read datasets. There are methods that can find split reads mapped to annotated genomes and can measure polymorphism. You have set up the demographic hypothesis as "lineages with higher N_e should experience more effective purifying selection in the removal (or prevent fixation) of longer... TE insertions" (lines 230-232). Therefore, it is a population-level phenomenon you are interested in and thus you should be looking at allele frequencies, and you did not explicitly test the hypothesis because you did not look at polymorphism.

And then, you are not really taking into account demography here - only species-wide estimates of N_e which are suspect to begin with. As shown in *Anolis* (and *Drosophila*), TE fixation rates and copy number can be drastically different between subpopulations. PSMC cannot identify population substructure. Also, it is not well-suited for inferring recent effective population sizes (see Li and Durbin 2011). The model is very sensitive to its assumptions, such as generation time and mutation rate. I am not sure why the authors decided to apply a generation time of 3 years to all eight species, as this is very likely to differ between the species - can this be justified? Also, with ~160 million years of divergence represented in the eight species, it is likely that the substitution rates would vary greatly, especially between the iguanian, anguimorph, and snakes, so I would like to see how this is justified as well. Perhaps you could corroborate the mutation rates used here with mutation rates estimated from another method, such as based on substitutions per site in pairwise alignments of your own sequence data given the divergence time: $(\text{pwise divergence}/2) / (\text{TMRCA}/\text{generation time for species})$. Another way would be to use adult body mass as a proxy for N_e - which is often done - see Figuet et al. 2016 MBE - but not without its caveats (again, lacks information about substructure). Or, use the mitochondrial data in an Extended Bayesian Skyline Plot to try to recover more recent estimates for N_e .

One thing going for PSMC is that it is good at modeling ancient demographic events (as long as the assumptions are sound and there is no substructure), and you do not use this information. For instance, the species with the most precipitous N_e expansion as demonstrated by the PSMC (*Thamnophis sirtalis*) also happens to have the most truncated elements. Also, a PSMC was not done on the most extensively studied genome assembly, *Anolis carolinensis*. I understand this was a Sanger-sequenced genome, but short reads should be available by now (Ruggiero et al 2017). It's unfortunate that the PSMC analysis of the genome wasn't done in order to corroborate the demographic explanation of TE dynamics that has been extensively studied in that genome with what the authors are trying to do here.

L119: "CR1s are particularly abundant and active" - what is your criteria for activity? do the elements differ very little from subfamily consensus which would suggest recent activity? Are they expressing a lot of reverse transcriptase?

L155: "mammal" should be "mammals"

L288: Why would you create separate libraries for snakes and lizards? Snakes are nested within lizards so I don't see why this dichotomy is applied.

Reviewer #2 (Remarks to the Author):

In their study titled "Squamate reptiles challenge paradigms of genomic repeat element evolution set by birds and mammals", Pasquesi et al. analyze in depth several characteristics of 66 squamate genomes with an evolutionary perspective and solid methods. Mainly, they explore the repeat dynamics (transposable elements and microsatellites), contrasting their results with what is known in mammals and birds. The quality of the manuscript as well as the scale and solidity of the new data and analyses presented make this work a great resource and formidable contribution to the fields of genomics, evolution and genome dynamics. Additionally, I believe that this manuscript will be of interest to a large audience, and thus should be warranted publication in Nature Communications. However, I have a couple of major (general) comments that affect the claims of this work and thus should be addressed prior to publication, complemented by detailed comments that I believe would increase the quality and clarity of the manuscript.

GENERAL (MAJOR) COMMENTS:

These general comments do not diminish the high quality and impact of this work, but the formulation of the claims in several instances in the manuscript needs to be revised and refocused (pointed in detailed comments).

- The authors are quite definitive mentioning an established paradigm of correlation between genome size and repeat content, but it is in fact more of an ongoing discussion, regularly tested with available data (thus, indeed biased). They did not cite a study that recently addressed this exact question (Elliott, T.A. and T.R. Gregory 2015 doi:10.1098/rstb.2014.0331), where the correlation for animals was weak despite being significant ($r=0.377$, $p=1.04 \times 10^{-4}$, $n=101$ contrasts), suggesting that it may not hold for subsets of animals. And indeed, based on the supp data from this study, the correlation does not hold considering only mammals or birds.
- While I agree that the link between genome size variation and transposable elements propagation and accumulation seem to differ between squamates and mammals or birds, the authors did not specifically test an "accordion" model type of genome size evolution since DNA loss or deletion rates are not measured. For example, deletion rates may be higher for squamate species with high recent TE activity than species with low recent TE activity, which would be compatible with this model (I do not believe that measuring CR1 truncation represents deletion rates). The text should be revised in several instances to reflect this, and the discussion modified accordingly. See detailed comments below.
- Why did the authors excluded the woodpecker from any TE range in birds? Do they have any evidence that it does not in fact contain as much TEs as previously published? Such exclusion should be clearly stated and explained, or the woodpecker should be considered in

generalizations about birds. See detailed comments below.

DETAILED COMMENTS:

ABSTRACT

- I26: the word defy is too strong, see comments below.
- I27: unparalleled is only true without considering the woodpecker.

INTRODUCTION

- I39-42: see general comment.
- I43: the cited study (about the "accordion" model) already highlights that the dynamics is different between mammals and birds, which supports the fact that studying genome size dynamics in other orders is crucial in order to gain a better understanding of the underlying mechanisms of the link between genome size and TEs. Indeed, in this study the net TE gains from the last 100 My correlates with assembly size in mammals, while total TE content does not (thus this can be seen as a refinement of the model that genome size correlates with TE content). In birds, the TE gains correlate with loss, supporting the "accordion" model.
- I51: with the supp data from Kapusta et al. 2017 (cited as the source for birds), even without the woodpecker this ratio is 2.4, not 2.2. Which values were used exactly? Additionally, with the woodpecker this ratio would be 5.4, which would end up being higher than the squamates (see also comments in the Results section). The same supp data for mammals gives a ratio of 2, and not 1.7; granted this is very similar, but which species / values were used in the calculations?
- I58: see general comment about the use of "contradicting".
- I73: see general comment about the 'accordion' model & detailed comments in the Results section.

RESULTS

- I96: see first general comment; the reference cited here is about TE diversity, and the reference Elliott, T.A. and T.R. Gregory 2015 (doi:10.1098/rstb.2014.0331) sounds more appropriate to me.
- I100: see comment about I51.
- I104: the max value for squamates is listed as 56.3, but should read 73.
- I105-111: here I would argue that what distinguishes squamates from birds and mammals would be TE contents in the range of the ones of mammals, but genome sizes comparable to the ones of birds rather than the variation in TE content (which, again, only holds because the woodpecker was excluded). It is indeed very interesting that the small and quite constrained genomes of squamates contain so large amounts of TEs, which I would emphasize more, instead of focusing on the ratios.
- I109: mentioning an example would be helpful, either in main text or in the figure legend (Coniophanes for sure; maybe Pantherophis, Ophisaurus...?). Also, Figure S3 is referred

here but it does not show variation within the same genus - Figure S2 seems more appropriate. However, then Figure S3 would not be referred to...?

- I123: the 2.4 fold is hard to visualize from the heat maps: a range in number added in the text would be helpful (also, refer to Table S2)

- I135 (and I143): the fact that *C. fissidens* has the highest GC content as well as the highest SSR content by far is quite interesting.

- I220: I do find the HT data quite compelling, but the other lines of evidence to conclude for HT are missing. Since the whole alignment is not provided, the reader cannot get an idea of how similar these BovB copies are (e.g. percentage of identity between the copies), and this could be contrasted to other regions of the genome. Additionally, do other ticks (if any) contain any BovB elements, or are they only found in these 2 ticks?

- I223: consider adding 'non exclusive', since the "accordion" model is expected to be affected as well by shifts in effective population size (would affect rates of fixation of deletions and insertions).

- I228-229: see also general comment. Genomic abundance and truncation of LINE elements is not an estimation of DNA loss, mostly since LINE truncation would (largely) happen upon integration.

- I234: The variations showed in Figure S10e are striking and very interesting - this is a very neat analysis, which emphasizes how complex it is to test the correlation between evolutionary dynamic features (TE content and N_e). This is where I would argue that the absence of correlation between the net amount of TEs and the current N_e values (e.g. lower N_e now for *P. vitticeps*) does not exclude that N_e variations influence(d) TE content (but more importantly, the fixation of non TE DNA deletions).

- I238: was a correlation expected? Please expand the reasoning.

- I240-244: the balance gain/loss was not directly tested for, so I would argue that the authors cannot completely exclude that squamate genomes do not fall under a gain/loss balance (maybe not as strong as birds, but in between birds and mammals?). Given their low variation in genome size, I would hypothesize that they do.

- I247 "...may instead explain variation in TE abundance": The "accordion" model is about genome size variations, not TE content variations (even though TE content is impacted).

- I248 "...given their apparent decoupling of genome size and TE abundance...": same comment as above (TE content does not correlate with genome size in birds in Kapusta et al. 2017).

- I249-252: also, DNA repair mechanisms. And the comments above do not invalidate this conclusion: more analyses to address these questions are definitely needed.

DISCUSSION

- I266-270: same comment as I247 and I248. The data presented here is not testing DNA loss and DNA gain coupling or de-coupling. Kapusta et al. only mentions the dynamics part of these models (e.g. Figure S4) so, indeed, more in depth studies based on large sample sizes are needed and squamates will likely be a formidable resource.

- I271: see comment I234.

MATERIAL

- Please consider providing the species tree used in the figures (in Newick or other format).

METHODS

- the version of BEDtools should be added when the tool is mentioned.
- I290: very nice masking strategy!

SUPP METHODS

- I49: what were the criteria to merge the TE libraries?
- I64-66: what does the customization involve?
- I75: please add some Methods for "We extracted at random subsamples". It sounds like reads were extracted, so how were 'RepeatMasker estimates' obtained?
- I85: Please include more details such as parameters or criteria for "mapping nuclear reads to the consensus of the most closely related species" (mostly since CLC genomic workbench is not a free software).
- I143: I am just curious, why remasking instead of using intersectBed for example, to interrogate existing RM annotation? Was this a way to select for really close TEs, since TEs not overlapping the 400 bp by at least 50bp or so would not be detected?
- I147: "genomic background reads were generating" should read "genomic background reads were generated".

FIGURES

GENERAL

- (small) thick marks on the My scale would be a nice addition

FIGURE1

- The violin plots are a great way to represent genome size data and these figures are very neat, and grouping species allows to plot genome size data for related species without TE content data; however, what is the exact date of access to the animal genome size database (only the year is provided, but the database is constantly updated) and the number of data points? This could a supplementary table.
- the heat scale goes up to 73 (C. fissidens TE content), yet the branch for C. fissidens is not red on Figure 1C like on Figure S2?
- The text says colubroids, but Figure 1C reads Colubrids?

FIGURE 4

- legend for C mentions left and right, should read up and down.

SUPPLEMENTARY INFORMATION

TABLE S1

- 4 species have 'Total Raw bp' < 'Total Nuclear bp'; how is this achieved?

TABLE S2

- it is not cited throughout the manuscript; should likely be added when Figure1 and S2 are

referred to.

- C. fissidens BovB content reads 302, likely a typo?

FIGURE S1C

- the colors do not match exactly between the lines and the legend.

FIGURE S2

- The figure would benefit from having the different genus names on the figure (could be rotated text which would limit the space issue), rather than different colors with no legend (it is easy to figure out from Figure 1, but it would help)
- consider adding in the legend examples making the point that is made in the text when this figure is cited, unless added in the text itself

FIGURE S3

- which point of the results does this figure support?
- how to read this figure is not very clear from the legend, unless the reader is already familiar with the Method employed. Consider adding one or two sentences/examples to guide interpretation.

FIGURE S4

- it would add to the quality of this analysis if the correlations were verified once phylogenetic relatedness is taken in account (phylogenetic contrasts). However, I do not think that this is a critical point to address.

FIGURE S9a

- Please place on the figure the complete species names of the ticks, since mentioned in the text
- Also, the species names are different here than in the other figures, please homogenize

FIGURES S13 & S14

- Consider expanding the legends to guide the reader on how to read these complex figures; such as lower n means more recently amplified copies (both figures) and so Scincoidea had recent CR1 amplifications (Figure S14A) and Anguimorpha recent BovB amplifications (Figure S14B).
- Please homogenize the names (e.g. Anguids everywhere else in the text and Figures, but Anguimorpha here).

Reviewer #1 (Remarks to the Author):

The authors have amassed a rather impressive genomic dataset from more than 60 squamates. They demonstrate that lower-coverage datasets can accurately predict genomic repeat content. They show that some squamate genomes are comprised of more SSRs than any studied vertebrate, and that snake genomes have been seeded with microsatellites by LINEs to an unprecedented degree. They also use phylogenetic analyses to demonstrate multiple horizontal transfer events (most of which has already been documented), including a previously unidentified instance of snake-to-tick BovB transfer. They then test two competing (although not necessarily mutually exclusive) hypotheses about the forces controlling TE diversity and abundance in vertebrate genomes - one where species demography affects the strength of purifying selection against deleterious elements and another where the balance between DNA gain and loss mediate the amount of repetitive DNA that can accumulate in the genome. While the intended scope of the manuscript is notable, I believe the authors have missed some opportunities to provide strong evidence for some of their conclusions, in particular that current hypotheses explaining repeat content variation across vertebrates do not hold for squamates.

*I am not wholly convinced that the lack of relationship between N_e and TE truncation as shown here rules out the role of demography in the accumulation success of TEs in a genome. For instance, BovB has been characterized in the *Anolis* genome, where it is very abundant and ancient (as calculated by average divergence from consensus) and likely extinct. Therefore, it is likely that these elements reached fixation in the *Anolis* genome long before the species evolved. Thus, they are probably more truncated because they are older and have been hit by more deletions over evolutionary time.*

To address the reviewer's concerns, we have added a number of new analyses to further test our hypotheses – the results of which all support our previous conclusions. First, to clarify, in our manuscript we hypothesize that TE truncation and effective population size (N_e) should be fundamentally related IF demography and the efficacy of purifying selection are the major drivers of TE proliferation in eukaryotic genomes – this hypothesis has been a major paradigm of TE biology to date. A major concern of the reviewer is that we did not address the alternative possibility that age of TEs (LINEs specifically) may determine TE truncation more than purifying selection and N_e .

As suggested by the reviewer, we have added new analyses to test for a correlation (or lack of) between TE age and truncation in two ways:

- 1) We compared truncation versus the median values of pi (divergence, proportional to age) across species (Fig. 4f).
- 2) We redid this same comparison in a phylogenetically aware framework (using Phylogenetically Independent Contrasts), based on suggestions for using these approaches by Reviewer 2 (Supplementary Fig. 13).

Both phylogenetically aware and unaware linear regressions are not significant, nor do the relationships appear compelling in any way. These new figures and relevant text pertaining to these figures have been added to the revised manuscript (Figure 4f and Supplementary Fig. 13).

We also note that the reviewer's comment about previous analyses in *Anolis* are based on biased estimates (for multiple reasons), which makes the expectation from analyses in *Anolis* less compelling. Previous estimates of BovB divergence

were based on what has since been identified as an erroneous (chimeric) BovB consensus sequence from RepBase (demonstrated by Castoe et. al., 2011, *GBE*), and previous divergence estimates did not account for subfamily structure by removing subfamily-defining sites (as we have done here). Thus, BovB estimates from previous studies are heavily biased towards ancient date estimates, yet once these biases are removed, it appears that BovB activity in *Anolis* is far more recent (e.g., Supplementary Fig. 15). Additionally, BovB elements show at least some recent activity in almost all squamate genomes, and full-length, potentially active BovB elements can indeed be detected in the *Anolis* genome (e.g., chr5 52563099-52566162, chr3 138977345-138974267, as suggested also by Ruggiero et al., 2017, *Front. Genetics*: “Finally, the RTE-BovB family contains a very small number of full-length elements, which is probably related to the fact that this family is on its way to extinction.”)

You didn't include green anole in the demographic analysis here, but let's assume Pogona as an iguanian is similar. Your plot in Figure 4 and in the supplementary shows this - more truncation in BovB than CR1 in Pogona. In contrast, the subfamilies of CR1 are currently producing copies in the Anolis genome (i.e., low average divergence from consensus), these may potentially produce new full length copies and thus would be susceptible to purifying selection. However, I am not sure why you lumped together all CR1 subfamily elements within each genome to calculate divergence - it's mentioned in the supplement that you excluded subfamily-diagnostic sites but wouldn't there also be random mutations in each subfamily that occurred over evolutionary time? To me, this is potentially problematic, and it should be relatively simple to break CR1 up into families and calculate divergence within each. Thus, the estimates of age and activity for element families may be biased towards older ages and lower activity. Please either explain better how you did this or try to calculate divergence within subfamilies. I wonder if similar age classes of subfamily show the same amount of truncation.

Indeed, the reviewer is correct that not separating elements by subfamily would provide highly biased results – this is not what we did. We apologize for not making our methods more clear in this section. As suggested by the reviewer, we did analyze divergence of each subfamily separately to avoid such biases and have made this much more clear in the main text and supplementary methods of the revised manuscript, and thank the reviewer for making certain that we clarify this point.

Also, the reviewer asks the question: “I wonder if similar age classes of subfamily show the same amount of truncation” – we have directly addressed this question by adding a new figure (revised Fig. 4f), which shows that very different trends of truncation are associated with subfamilies of similar ages. This is relevant because it counters the concern of the reviewer that patterns of truncation might be explained by being auto-correlated with age – these new figures show that this is not the case and provide added evidence for fundamentally different processes leading to different patterns of truncation in different genomes.

Perhaps more importantly, the amount of 3':5' truncation wouldn't necessarily be an indication of the strength of purifying selection acting against elements - target primed reverse transcription is mistake-prone and more often than not leaves truncated copies, so we expect more truncation in any genome regardless of demographic history. Meanwhile, purifying selection is a post-insertional force. ...

While the reviewer is certainly correct about the biology of retroelements, we disagree about key details of their assumptions. For example, while 5' truncation is indeed a natural pattern associated with all retroelements (due to the lack of processivity of reverse-transcriptase), this bias creates a distribution of new elements of varying lengths. Therefore, while we certainly expect truncation in any genome for all retroelements, we are focusing on the degree of truncation observed in these elements, which may vary due to purifying selection (e.g., differential fixation of more versus less truncated copies). For example, whether or not these elements are fixed, and whether these elements are retained or deleted from genomes over evolutionary time may indeed depend on purifying selection. Therefore, purifying selection may act on multiple steps in this process, including insertional and post-insertional phases.

... Therefore, it would be more meaningful to look at the level of fixation in the genome - something entirely possible with your short read datasets. There are methods that can find split reads mapped to annotated genomes and can measure polymorphism. You have set up the demographic hypothesis as "lineages with higher N_e should experience more effective purifying selection in the removal (or prevent fixation) of longer... TE insertions" (lines 230-232). Therefore, it is a population-level phenomenon you are interested in and thus you should be looking at allele frequencies, and you did not explicitly test the hypothesis because you did not look at polymorphism.

The approaches suggested by the reviewer (identifying polymorphic repeat alleles and measuring their frequencies) would not actually be possible given the nature of our data (or the nature of the analysis). The majority of our data are unassembled, low-coverage sampled genomes from a *single representative*; the remainder of our data are assembled genomes from a *single individual*. None of these data include sampling from *multiple individuals*, which would be necessary for obtaining meaningful estimates of allele frequencies associated with TE polymorphisms. We have recently published results based on the approaches suggested by the reviewer (Ruggiero et al, 2017), and know that unfortunately these will not work effectively on the data generated for this study.

While fixation of elements is a population-level phenomenon, as the reviewer points out above, fixation can be recent or ancient. Either way, the process of fixation will result in analyzable patterns of TEs fixed in modern-day genomes. In fact, this is argued particularly well by Lynch and Conery in their original thesis for the link between N_e and genome size (Lynch and Conery, 2003). Therefore, because we can analyze the results of fixed TEs we can indeed infer changes along a tree that are either consistent or not with hypotheses.

*And then, you are not really taking into account demography here - only species-wide estimates of N_e which are suspect to begin with. As shown in *Anolis* (and *Drosophila*), TE fixation rates and copy number can be drastically different between subpopulations. PSMC cannot identify population substructure. Also, it is not well-suited for inferring recent effective population sizes (see Li and Durbin 2011). The model is very sensitive to its assumptions, such as generation time and mutation rate. I am not sure why the authors decided to apply a generation time of 3 years to all eight species, as this is very likely to differ between the species - can this be justified? Also, with ~160 million years of divergence represented in the eight species, it is likely that the*

substitution rates would vary greatly, especially between the iguanian, anguimorph, and snakes, so I would like to see how this is justified as well. Perhaps you could corroborate the mutation rates used here with mutation rates estimated from another method, such as based on substitutions per site in pairwise alignments of your own sequence data given the divergence time: (pwise divergence/2) / (TMRCAs/generation time for species). Another way would be to use adult body mass as a proxy for N_e - which is often done - see Figuet et al. 2016 MBE - but not without its caveats (again, lacks information about substructure). Or, use the mitochondrial data in an Extended Bayesian Skyline Plot to try to recover more recent estimates for N_e .

This was an excellent suggestion by the reviewer and now provides a secondary line of inference that further tests our hypothesis (and provides confirmation of our previous conclusions). In an effort to provide additional estimates of N_e that are independent of demographic and mutation rate, we have added new analyses (and associated text) that use adult body mass (Supplementary Table 9) as a proxy for N_e . We use these alternative inferences of relative N_e , together with our previous estimates, to provide additional comparisons of truncation patterns that test the hypothesis that N_e and truncation are linked (Fig 4e and Supplementary Fig. 12). We have also added relevant text identifying the added power and confirmation that both approaches for N_e estimation provide.

The reviewers' concern about substructure biasing PSMC estimates is not relevant here because all PSMC analyses are based on a single individual (i.e., substructure is inherently not relevant). Also, these demography estimates are thus derived from the same individuals we analyzed for TE content so as much as is possible, we controlled for this in the context of our analyses.

For our PSMC estimates, we used the generalized squamate mutation rate reported in Green et al. 2014 (and this is cited accordingly). Further, Green et al. (2014, *Science*), as well as Castoe et al. (2013, *PNAS*), both provide evidence of relatively similar mutation rates across lineages of squamates, and this was our justification for using a single rate for all species – this is also now discussed and cited in the Supplementary Methods. Also, for our PSMC estimates, we did apply a generation time of 3 years across the 8 species because this was the average of generation time approximations we retrieved from the literature (Supplementary Table 11) – the origins and logic of these estimates are now explicitly cited and explained in the Supplemental Methods. Further, to demonstrate that reasonable variation in generation times have little-to-no effect on our inferences, we also re-ran PSMC with alternative generation times within the range reported in the literature, and our alternative runs of PSMC yielded remarkably similar estimates of average N_e but with slightly different temporal estimates for population growth and decline – because these were so similar to the figures shown in the main text, we did not include these in the revised supplemental information.

Although the reviewer suggested the possibility of deriving demographic inferences also from mitochondrial data using Bayesian skyline plots, such estimates would require population sampling from many individuals per species for mitochondrial data, which are not available.

One thing going for PSMC is that it is good at modeling ancient demographic events (as long as the assumptions are sound and there is no substructure), and you do not use this information. For instance, the species with the most precipitous Ne expansion as demonstrated by the PSMC (Thamnophis sirtalis) also happens to have the most truncated elements. Also, a PSMC was not done on the most extensively studied genome assembly, Anolis carolinensis. I understand this was a Sanger-sequenced genome, but short reads should be available by now (Ruggiero et al 2017). It's unfortunate that the PSMC analysis of the genome wasn't done in order to corroborate the demographic explanation of TE dynamics that has been extensively studied in that genome with what the authors are trying to do here.

The highest-coverage short read dataset available for *Anolis* is 13x. Recent studies have strongly suggested that PSMC should not be run on datasets with <18x coverage (Nadachowska-Brzyska et al., 2016, Mol. Ecol). In our own experience with datasets in the range of 15-20x coverage, even if PSMC does run, it often returns nonsense results (or no results at all). For these reasons, we did not run PSMC on *Anolis*. Also, even if we were able to run PSMC on *Anolis*, we would have the complication that this demographic inference would not be from the same population as the genome sequence, which would make the comparison between demography inferred from PSMC and TEs in the *Anolis* genome less reasonable.

L119: "CR1s are particularly abundant and active" - what is your criteria for activity? do the elements differ very little from subfamily consensus which would suggest recent activity? Are they expressing a lot of reverse transcriptase?

We have expanded text here to be more clear that our criteria for recent activity is very low sequence divergence (π) of many copies of CR1 elements (i.e., many elements differ very little from subfamily consensus sequences). To further clarify this statement, we have also cited Supplementary Fig. 15 that shows distributions of π for CR1 for all species. None of our data provide any resolution regarding the expression level of reverse transcriptase (since we have no RNAseq-based data).

L156: "mammal" should be "mammals"

This change has been made.

L288: Why would you create separate libraries for snakes and lizards? Snakes are nested within lizards so I don't see why this dichotomy is applied.

To clarify the rationale for our approach we have added brief discussion and justification to the revised supplement, including a new figure (Supplementary Fig. 17). The standard in the field of repeat annotation is to use a single species-derived repeat library, but this approach limits detection power, and may create biases when comparing assembled versus unassembled or less complete genomes. To provide greater power and reduce biases in repeat identification, we made 2 large libraries for repeat masking. The reason we made 2 (rather than 1) was to reduce the repeat masking time (by reducing overall library size). We also found that this resulted in no significant change in the masking results. To confirm that this is a reasonable approach, we have provided a new figure in the supplement to demonstrate that masking with a combined library versus a

separate snake or lizard specific library results in no notable change in inferred TE content and most importantly, in the amount of repeats identified.

Reviewer #2 (Remarks to the Author):

In their study titled "Squamate reptiles challenge paradigms of genomic repeat element evolution set by birds and mammals", Pasquesi et al. analyze in depth several characteristics of 66 squamate genomes with an evolutionary perspective and solid methods. Mainly, they explore the repeat dynamics (transposable elements and microsatellites), contrasting their results with what is known in mammals and birds. The quality of the manuscript as well as the scale and solidity of the new data and analyses presented make this work a great resource and formidable contribution to the fields of genomics, evolution and genome dynamics. Additionally, I believe that this manuscript will be of interest to a large audience, and thus should be warranted publication in Nature Communications. However, I have a couple of major (general) comments that affect the claims of this work and thus should be addressed prior to publication, complemented by detailed comments that I believe would increase the quality and clarity of the manuscript.

We are pleased about the enthusiasm of the reviewer for the paper, and thank them for the many excellent points and suggestions they provide below. As we outline in detail below, we have done our best to fully address these points, concerns, and suggestions in the revision which we believe has substantially improved the overall impact and quality of the manuscript and the clarity of key points.

GENERAL (MAJOR) COMMENTS:

These general comments do not diminish the high quality and impact of this work, but the formulation of the claims in several instances in the manuscript needs to be revised and refocused (pointed in detailed comments).

- The authors are quite definitive mentioning an established paradigm of correlation between genome size and repeat content, but it is in fact more of an ongoing discussion, regularly tested with available data (thus, indeed biased). They did not cite a study that recently addressed this exact question (Elliott, T.A. and T.R. Gregory 2015 doi:10.1098/rstb.2014.0331), where the correlation for animals was weak despite being significant ($r=0.377$, $p=1.04 \times 10^{-4}$, $n=101$ contrasts), suggesting that it may not hold for subsets of animals. And indeed, based on the supp data from this study, the correlation does not hold considering only mammals or birds.

We have revised the text in this section in an effort to reduce the definitive nature of our depiction of the literature, and have added the suggested citation in this section (as well as in the relevant part of the revised discussion).

- While I agree that the link between genome size variation and transposable elements propagation and accumulation seem to differ between squamates and mammals or birds, the authors did not specifically test an "accordion" model type of genome size evolution since DNA loss or deletion rates are not measured. For example, deletion rates may be higher for squamate species with high recent TE activity than species with low recent TE activity, which would be

compatible with this model (I do not believe that measuring CRI truncation represents deletion rates). The text should be revised in several instances to reflect this, and the discussion modified accordingly. See detailed comments below.

The reviewer raises a valuable point that the type of data and analyses in this paper do not directly test the accordion model and whether this model is indeed underlying patterns of TE diversity and abundance in squamates. Throughout the revised manuscript we have modified the text to maintain the mention of this recently proposed model, but specifically avoid confusing this with the notion that we are testing it, or its predictions. Many of these changes are brought up below by specific Reviewer 2 comments, where we provide the details of how they were addressed. Most relevant to the citations of the accordion model, we have substantially revised the last paragraphs of the Results and Discussion to clarify the relevance of our findings to the accordion model.

- Why did the authors excluded the woodpecker from any TE range in birds? Do they have any evidence that it does not in fact contain as much TEs as previously published? Such exclusion should be clearly stated and explained, or the woodpecker should be considered in generalizations about birds. See detailed comments below.

We have added explanation to the main text (as well as to the supplement) to make this more clear and transparent, as requested by the reviewer – please see comments below for details.

DETAILED COMMENTS:

ABSTRACT

- 126: the word defy is too strong, see comments below.

We have changed this.

- 127: unparalleled is only true without considering the woodpecker.

After careful consideration, we have kept the current wording for two reasons. First, even including the woodpecker in the bird estimate of repeat variation, the squamate variation is higher (highest) – please see below for more detail. Second, the average variation between any two squamate species is quite extreme and unparalleled.

INTRODUCTION

- 139-42: see general comment.

- 143: the cited study (about the "accordion" model) already highlights that the dynamics is different between mammals and birds, which supports the fact that studying genome size dynamics in other orders is crucial in order to gain a better understanding of the underlying mechanisms of the link between genome size and TEs. Indeed, in this study the net TE gains from the last 100 My correlates with assembly size in mammals, while total TE content does not (thus this can be seen as a refinement of the model that genome size correlates with TE content). In birds, the TE gains correlate with loss, supporting the "accordion" model.

This is an excellent point. In the spirit of the reviewer's comment we have modified this section to point out that this variation in model/pattern across lineages has been noted previously, and that such variation highlights the value of analyses of diverse vertebrate lineages.

- 151: with the supp data from Kapusta et al. 2017 (cited as the source for birds), even without the woodpecker this ratio is 2.4, not 2.2. Which values were used exactly?

Additionally, with the woodpecker this ratio would be 5.4, which would end up being higher than the squamates (see also comments in the Results section).

The reviewer's values do not match that provided in Kapusta. According to the Supplementary Data S2 of Kapusta et al. 2017 for 28 bird genomes, the authors report a ratio of 4.8 including the woodpecker, and of 2.16 without the woodpecker (which we rounded up to 2.2).

These values in birds, even including the woodpecker, are still less than that observed in squamates, and for this reason we have maintained several claims related to squamates being the most extreme in terms of repeat content variation.

The same supp data for mammals gives a ratio of 2, and not 1.7; granted this is very similar, but which species / values were used in the calculations?

For the estimates of repeat content of mammals, we used data available from the RepeatMasker GenomicDatasets web page (<http://www.repeatmasker.org/genomicDatasets/RMGenomicDatasets.html>) and original genome papers that specifically used species-specific de-novo TE annotations (because otherwise, their estimates are strongly biased to be low). To be more clear about the origins of these calculations and estimates, we have added the corresponding data and sources to Supplementary Table 1. Based on these data, the resulting ratio for mammals is 1.68, which we rounded up to 1.7.

- 158: see general comment about the use of "contradicting".

We have deleted the second half of this sentence altogether (including the part that discusses the "contradiction of the paradigm...")

- 173: see general comment about the 'accordion' model & detailed comments in the Results section.

We deleted this sentence that mentioned the accordion model here because, as described above, we agree with the reviewer that we did not formally evaluate this model.

RESULTS

- 196: see first general comment; the reference cited here is about TE diversity, and the reference Elliott, T.A. and T.R. Gregory 2015 (doi:10.1098/rstb.2014.0331) sounds more appropriate to me.

We agree and have changed the citation accordingly.

- 1100: see comment about 151.

This comment relates to the mention of the woodpecker, and that fact that we omitted this sample from some of our estimates. We have added text to this section (as well as to the supplement) to make this more clear and transparent, as requested by the reviewer.

- 1104: *the max value for squamates is listed as 56.3, but should read 73.*

The reviewer must have slightly misread this section. Here, to be consistent with our comparisons to birds and mammals, we are referring just to elements identified as TEs specifically, not total repeat content, so this number is indeed correct (but we thank the reviewer for being so diligent as to notice).

- 1105-111: *here I would argue that what distinguishes squamates from birds and mammals would be TE contents in the range of the ones of mammals, but genome sizes comparable to the ones of birds rather than the variation in TE content (which, again, only holds because the woodpecker was excluded). It is indeed very interesting that the small and quite constrained genomes of squamates contain so large amounts of TEs, which I would emphasize more, instead of focusing on the ratios.*

As discussed above, even with the woodpecker included, the variation is greatest within squamates. Otherwise, we agree with the reviewer and have added a sentence partially paraphrasing the point of the reviewer: “our results highlight the remarkable finding that the comparatively small genomes of squamates can contain large amounts and highly variable amounts of repeat elements”

- 1109: *mentioning an example would be helpful, either in main text or in the figure legend (Coniophanes for sure; maybe Pantherophis, Ophisaurus...?).*

We agree and have added several examples here.

Also, Figure S3 is referred here but it does not show variation within the same genus - Figure S2 seems more appropriate. However, then Figure S3 would not be referred to...?

The reviewer is correct, and we apologize for this error and thank the reviewer for bringing it to our attention. We changed the text to refer to the correct figures (e.g., Figures 1C and S2), and we have added a citation for Fig. S3 elsewhere in the text where it was appropriate.

- 1125: *the 2.4 fold is hard to visualize from the heat maps: a range in number added in the text would be helpful (also, refer to Table S2)*

We added the average values for colubroid snakes and lizards (from which the ratio was estimated), and also added a reference Table S2 that contains the raw data.

- 1135 (and 1143): *the fact that C. fissidens has the highest GC content as well as the highest SSR content by far is quite interesting.*

Indeed, we also found this exciting and potentially suggestive of a link between SSR content and other processes, although we do not possess the data here to test such speculation.

- 1220: I do find the HT data quite compelling, but the other lines of evidence to conclude for HT are missing. Since the whole alignment is not provided, the reader cannot get an idea of how similar these BovB copies are (e.g. percentage of identity between the copies), and this could be contrasted to other regions of the genome.

To address the reviewer's request, we have added the CR1 and BovB alignments and trees as additional datasets (Supplementary Data 1 and 2).

Additionally, do other ticks (if any) contain any BovB elements, or are they only found in these 2 ticks?

These are the only two BovB tick hits that we are aware of based on Blast Searches of NCBI.

- 1223: consider adding 'non exclusive', since the "accordion" model is expected to be affected as well by shifts in effective population size (would affect rates of fixation of deletions and insertions).

We agree and have added 'non exclusive' as correctly suggested by the reviewer.

- 1228-229: see also general comment. Genomic abundance and truncation of LINE elements is not an estimation of DNA loss, mostly since LINE truncation would (largely) happen upon integration.

The reviewer is correct in that truncation is not a direct measurement of DNA loss. We have substantially revised the MS throughout (and in this section) to remove the accordion model from aspects of the text where we refer to testing hypotheses, and here have deleted an entire paragraph that contained the text with issues brought up by the reviewer.

*- 1234: The variations showed in Figure S10e are striking and very interesting - this is a very neat analysis, which emphasizes how complex it is to test the correlation between evolutionary dynamic features (TE content and N_e). This is where I would argue that the absence of correlation between the net amount of TEs and the current N_e values (e.g. lower N_e now for *P vitticeps*) does not exclude that N_e variations influence(d) TE content (but more importantly, the fixation of non TE DNA deletions).*

This is an insightful point that we are happy the reviewer brought up (or really, 2 points). First, we have added the point made by the reviewer about the complexity of trying to understand correlations between two such dynamic processes as N_e and TE content towards the end of the revised Results section. The second point – that N_e could still influence other aspects of genome evolution, like deletion fixation, that would indirectly impact TE content and be relevant to the accordion model and the balancing of genome size – has been added to the Discussion (middle of first paragraph).

- 1238: *was a correlation expected? Please expand the reasoning.*

This was confusing and we decided to reword the sentence to have a different, and more clear meaning.

- 1240-244: *the balance gain/loss was not directly tested for, so I would argue that the authors cannot completely exclude that squamate genomes do not fall under a gain/loss balance (maybe not as strong as birds, but in between birds and mammals?). Given their low variation in genome size, I would hypothesize that they do.*

We agree, and as mentioned above, we have substantially revised all references to the accordion model (i.e., removed it from all aspects of text that deal with hypothesis testing), which has resulted in it being removed from this section – thereby fixing the issue brought up by the reviewer.

- 1247 *"...may instead explain variation in TE abundance": The "accordion" model is about genome size variations, not TE content variations (even though TE content is impacted).*

Agreed, and removed from this region of the MS.

- 1248 *"...given their apparent decoupling of genome size and TE abundance...": same comment as above (TE content does not correlate with genome size in birds in Kapusta et al. 2017).*

We agree, and as mentioned above, we have substantially revised all references to the accordion model (i.e., removed it from all aspects of text that deal with hypothesis testing), which has resulted in it being removed from this section – thereby fixing the issue brought up by the reviewer.

- 1249-252: *also, DNA repair mechanisms. And the comments above do not invalidate this conclusion: more analyses to address these questions are definitely needed.*

We agree and have added DNA repair mechanisms to this sentence, and added an additional sentence stating the need for future studies that examined such mechanistic explanations.

Also, we note that we have heavily revised the end of the Results and Discussion, which has resulted in essentially moving and merging this previous section into the last paragraph of the new Discussion (where it seems more appropriate).

DISCUSSION

- 1266-270: *same comment as 1247 and 1248. The data presented here is not testing DNA loss and DNA gain coupling or de-coupling. Kapusta et al. only mentions the dynamics part of these models (e.g. Figure S4) so, indeed, more in depth studies based on large sample sizes are needed and squamates will likely be a formidable resource.*

We have substantially re-written the Discussion and believe that this issue has been addressed throughout the revised MS. The original sentence flagged by the reviewer was deleted and replaced completely as part of this Discussion re-write.

- 1271: see comment 1234 > “in Figure S10e ... This is where I would argue that the absence of correlation between the net amount of TEs and the current N_e values (e.g. lower N_e now for *P vitticeps*) does not exclude that N_e variations influence(d) TE content (but more importantly, the fixation of non TE DNA deletions).”

Please see comments above describing how we have addressed this in the revised MS.

MATERIAL

- Please consider providing the species tree used in the figures (in Newick or other format).

The revised manuscript now includes the newick tree files as supplementary data files (Supplementary Dataset 1 for BovB and Supplementary Dataset 2 for CR1-L3).

METHODS

- the version of BEDtools should be added when the tool is mentioned.

We have added the suggested citation to the Bedtools version used in the supplemental methods

- 1290: very nice masking strategy!

We are pleased that the reviewer appreciates the substantial effort it required to conduct this masking strategy.

SUPP METHODS

- 149: what were the criteria to merge the TE libraries?

We provided more detailed information in the supplementary methods regarding the library generation and merging, as suggested by the reviewer

- 164-66: what does the customization involve?

We added summary information in the .tab output file for TE subfamilies of relevant importance for squamate reptiles, especially considering that according to the official ProcessRepeat script, Penelope elements are still classified as LINEs. We corrected for this, and included the subdivision of CR1-like LINEs, as well as other families, according to the classification provided in Chalopin et al. 2015 (GBE). These details have been added to the Supplementary Methods.

- 175: please add some Methods for "We extracted at random subsamples". It sounds like reads were extracted, so how were 'RepeatMasker estimates' obtained?

We added more details and related references in the Supplementary Information.

- 185: Please include more details such as parameters or criteria for "mapping nuclear reads to the consensus of the most closely related species" (mostly since CLC genomic workbench is not a free software).

We added additional details about this.

- 1143: *I am just curious, why remasking instead of using intersectBed for example, to interrogate existing RM annotation? Was this a way to select for really close TEs, since TEs not overlapping the 400 bp by at least 50bp or so would not be detected?*

The basic reasoning was two-fold: simplicity and specificity... We first masked our genomes for Simple Repeats only to increase masking specificity (since some of our consensus sequences contained TEs with SSR tails and we wanted to mask these two types of elements separately). Then, we extracted the genomic location of AATAG and ATAG loci, went 400bp upstream/downstream, and masked again for only TEs, to again increase masking specificity. This strategy also allowed us to better identify really close TEs, as correctly inferred by the reviewer. Additional details were added in text to clarify these points.

- 1147: *"genomic background reads were generating" should read "genomic background reads were generated".*

We thank the reviewer for catching our mistake, and changed it accordingly.

FIGURES

GENERAL

- (small) thick marks on the My scale would be a nice addition

We added the marks as suggested by the reviewer in the time scale of all Figures.

FIGURE 1

- *The violin plots are a great way to represent genome size data and these figures are very neat, and grouping species allows to plot genome size data for related species without TE content data; however, what is the exact date of access to the animal genome size database (only the year is provided, but the database is constantly updated) and the number of data points? This could be a supplementary table.*

We added the date we accessed the animal genome size database and the complete dataset used to generate the plots for mammals, birds and squamates in Supplementary Table 2. TE abundance data and sources are also available in Supplementary Table 1.

- *the heat scale goes up to 73 (C. fissidens TE content), yet the branch for C. fissidens is not red on Figure 1C like on Figure S2?*

We thank the reviewer for their diligence and attention to detail, but these figures and colors are correct as is. The reason for the confusion is that one tree (Figure S2) is depicting total repeat content (which is 73 for *C. fissidens*), while the other tree (Figure 1C) is depicting total TE content (which for *C. fissidens* is 56%).

- *The text says colubroids, but Figure 1C reads Colubrids?*

This is correct because while “colubrid” refers to the family Colubridae, Colubroidea actually refers to a much larger clade that includes vipers, elapids,

colubrids, and other snakes. To help make this more clear, we have modified figure 1 (and Supplementary Fig. 2) to include both of these names, thus showing the contents on the tree of the Colubridae vs the Colubroidea.

FIGURE 4

- legend for C mentions left and right, should read up and down.

We thank the reviewer for noticing the incongruence, and indeed modified the caption so that it now reflects the revised version of Figure 4.

SUPPLEMENTARY INFORMATION

TABLE S1

- 4 species have 'Total Raw bp' < 'Total Nuclear bp'; how is this achieved?

We thank the reviewer for noticing what was obviously a mistake on our part in reporting the numbers. (Now Supplementary Table S3)

TABLE S2

- it is not cited throughout the manuscript; should likely be added when Figure 1 and S2 are referred to.

The citation to the table has been added (Supplementary Table S4) as suggested by the reviewer.

- C. fissidens BovB content reads 302, likely a typo?

We thank the reviewer for noticing the clear typo, and corrected the value to 3.02 in the revised Supplementary Table S4

FIGURE S1C

- the colors do not match exactly between the lines and the legend.

We thank the reviewer, and modified the colors of the legend to match the plot.

FIGURE S2

- The figure would benefit from having the different genus names on the figure (could be rotated text which would limit the space issue), rather than different colors with no legend (it is easy to figure out from Figure 1, but it would help)

We modified the figure according to the suggestion of the reviewer.

- consider adding in the legend examples making the point that is made in the text when this figure is cited, unless added in the text itself

We thank the reviewer for the valuable suggestion, and added specific examples in the revised version of the main text, and in the figure caption.

FIGURE S3

- which point of the results does this figure support?

The figure shows the results of (phylogenetically aware) testing for differential evolutionary rates of the genomic repeat element landscape, and in particular shows how different lineages underwent significant differential expansion of specific TEs (colubroid snakes and geckos in particular). A specific reference to this figure has been added in the revised version of the manuscript.

- how to read this figure is not very clear from the legend, unless the reader is already familiar with the Method employed. Consider adding one or two sentences/examples to guide interpretation.

We added additional information on how to read the figure in the revised version of the supplementary figures.

FIGURE S4

- it would add to the quality of this analysis if the correlations were verified once phylogenetic relatedness is taken in account (phylogenetic contrasts). However, I do not think that this is a critical point to address.

This is a good suggestion on the part of the reviewer, and we have added phylogenetically independent contrasts here and elsewhere in the MS, including Supplementary Fig. 10-13.

FIGURE S9a

- Please place on the figure the complete species names of the ticks, since mentioned in the text

We have remade this figure but replaced all names with latin names to improve the reference-ability.

- Also, the species names are different here than in the other figures, please homogenize

Here and throughout, we have gone through figures to be certain that we homogenized the names used, and thank the reviewer for the suggestion.

FIGURES S13 & S14

- Consider expanding the legends to guide the reader on how to read these complex figures; such as lower π means more recently amplified copies (both figures) and so Scincoidea had recent CR1 amplifications (Figure S14A) and Anguimorpha recent BovB amplifications (Figure S14B).

We modified the captions as suggested by the reviewer (now Supplementary Fig. 15 and 16).

- Please homogenize the names (e.g. Anguids everywhere else in the text and Figures, but Anguimorpha here).

We homogenized the names used to refer to clades throughout the figures and text, as suggested by the reviewer.

Reviewers' comments:

Reviewer #1 (Remarks to the Author):

The authors responded to many of the criticisms and suggestions of the original submission, but I still think some reviewer concerns have not been adequately addressed. In particular, their test of the demographic explanation for repeat variation which is a major part of their manuscript needs to be either upgraded with additional tests as explained below, or its conclusions curtailed in importance as they are based on too many assumptions about the cause of deleterious mutations and the underlying demographic models.

Despite the author's response that critiques of PSMC - namely that it cannot capture substructure - are not relevant, these critiques have been previously described in detail and they need to be taken into account. Key conclusions of two studies which the authors cite both in their manuscript and in their rebuttal (Li & Durbin 2011; Nadachowska-Brzyska et al. 2016) explain the important caveats associated with the use of PSMC to apply species-wide models of N_e when there is ancient population structure. For instance, if introgression occurred between ancient subpopulations then the average coalescence of alleles would be larger (by introducing alleles with coalescence time prior to the divergence of the two subpopulations) and inflate the estimate of N_e .

The other limitation is the fact that more recent ($\sim 20,000$ years before present for human) and more ancient (~ 3 million years before present for human) N_e estimates are unreliable for PSMC (Li & Durbin 2011). The authors use median N_e across the whole PSMC for each species, although typically, the earliest bins are removed from analysis.

I don't want to stop them from using the PSMC - it is a fantastic tool - but if they are going to derive their demographic models from single diploid genomes, then the authors should include a discussion of these caveats, at least if they are to rely so heavily on the N_e estimates of the PSMC.

The authors focus on the degree of truncation observed in CR1 and BovB elements across squamate genomes, with the expectation that a greater degree of truncation in the genome would be due to purifying selection. This is an essential assumption of their test of a demographic explanation for variation of repeat content, which they ultimately conclude has "poor explanatory power" for squamate genomes in general. However, their test of the demographic model was critiqued in the previous review, and in my opinion the authors have not adequately addressed this important reviewer concern.

I agree with the author response in that purifying selection COULD act against longer elements and if so COULD result in fewer full-length and more truncated copies - this is predicted by an ectopic recombination model where the strength of selection is related to element length (Petrov et al MBE 2003). But the authors do not explain this at all and their reasoning - if it is along those lines - should be more explicitly stated. Why do they think more truncation is indicative of purifying selection?

Other models exist that explain the nature of purifying selection against TEs (such as gene disruption or toxic transposition), with their own sets of predictions - see Barron et al. *Annu Rev Genet* 2014. Perhaps their data could be used to test these hypotheses. I won't suggest experiments, but I want to highlight that if the authors want to test the demographic model by using truncation as a measure of purifying selection they need to first establish that purifying selection is (or is not) related to element length. Examining a genome assembly by looking at the length and age distributions of truncations can generate the hypothesis that purifying selection may or may not be controlling element diversity - but it still needs to be tested. In the literature, the most convincing tests come from examinations of polymorphism (Petrov et al *MBE* 2003, Neafsey et al *MBE* 2004, Blass et al. *GBE* 2012, Tollis et al. *GBE* 2013). I understand that the authors only generated short read data from single individuals, but this should not preclude the ability to detect polymorphic insertions using split reads (case in point: they used the same data to detect polymorphisms in single individuals with the PSMC). It was a suggestion made in the previous review, and the authors decided not to apply it. Overall, their test of the demographic explanation lacks power - rather than the demographic explanation itself.

Neafsey et al (2004) and others have shown that the cause of the deleterious effects of TEs may be very different across taxa, and that certainly may be the case over 200 million years of squamate evolution. The authors invoke this in their discussion and also suggest host regulation may differ between squamate lineages. They provide a tantalizing look into the genome structure of one of the most diverse and enigmatic groups of vertebrates - but ultimately I think the demographic model is inadequately tested here. These shortcomings need to be addressed - either by more explicitly describing the caveats in their demographic models and their proxy for purifying selection and recalibrating the conclusions in proportion to the inherent weaknesses of those assumptions, or by adding more experiments.

Reviewer #2 (Remarks to the Author):

I thank the authors for revising their manuscript by addressing both reviewers' comments. I find these revisions very satisfying and believe that they greatly improved the quality of this work, and thus believe that it should be warranted publication in *Nature Communications* without additional revisions.

Reviewer #1 (Remarks to the Author):

The authors responded to many of the criticisms and suggestions of the original submission, but I still think some reviewer concerns have not been adequately addressed. In particular, their test of the demographic explanation for repeat variation which is a major part of their manuscript needs to be either upgraded with additional tests as explained below, or its conclusions curtailed in importance as they are based on too many assumptions about the cause of deleterious mutations and the underlying demographic models.

Despite the author's response that critiques of PSMC - namely that it cannot capture substructure - are not relevant, these critiques have been previously described in detail and they need to be taken into account. Key conclusions of two studies which the authors cite both in their manuscript and in their rebuttal (Li & Durbin 2011; Nadachowska-Brzyska et al. 2016) explain the important caveats associated with the use of PSMC to apply species-wide models of N_e when there is ancient population structure. For instance, if introgression occurred between ancient subpopulations then the average coalescence of alleles would be larger (by introducing alleles with coalescence time prior to the divergence of the two subpopulations) and inflate the estimate of N_e .

We apologize for not dealing with this concern more directly in the previous review – we did not interpret the reviewer's previous comment/concern about 'population structure' in PSMC to include reference to ancestral introgression/hybridization. As the reviewer knows, there is no straightforward way to correct for possible biases associated with potential ancestral structure, and with only a single sampled genome per species, there is no way for us to assess if such ancestral structure may exist in our samples. Thus, the only ways around this potential caveat of the method were to: 1) use an alternative method to approximate N_e – and this is why we also tested for correlations between demography and TE characteristics using body mass as an independent proxy for N_e (as suggested by R1 in the previous review), and 2) add additional caveat statements to both the manuscript and supplementary methods, which we have done here in the second revised version. Please see our more detailed comments below in which we discuss further how we have addressed these caveats of the method in the revised manuscript.

The other limitation is the fact that more recent (~20,000 years before present for human) and more ancient (~3million years before present for human) N_e estimates are unreliable for PSMC (Li & Durbin 2011). The authors use median N_e across the whole PSMC for each species, although typically, the earliest bins are removed from analysis.

We actually did exclude the most recent and ancient estimates of N_e from our PSMC analyses, but we failed to make this sufficiently clear, and agree with the reviewer that it is important to do so (and thank them for pointing out the lack of clarity in our previous text). We have modified the Methods and Supplementary Methods to more clearly point out that we did, indeed, exclude the first and last time points from each PSMC analysis prior to any comparisons or calculations of median values, and we have also added appropriate citations justifying this approach (Li and Durbin 2011, Nielsen and Beaumont 2009, Mazet et al 2015, Boitard et al 2016, Nadachowska-Brzyska et al. 2016 ...).

To further verify that other alternative adjustments/filtering of N_e estimates also do not change broad conclusions about correlations with N_e and TE characteristics, we conducted additional experiments with further filtering of time points for estimating median N_e from PSMC:

- i) Including only time points within 20,000 – 10,000,000 YBP

- ii) excluding ~10% more extreme time points (points: 1, 27-28)
- iii) excluding ~25% more extreme time points (points: 1, 24-28)

As with our previous analyses, none of these alternative filtering schemes resulted in a significant correlation between median N_e and CR1/BovB truncation, nor between median N_e and total TE or total repeat element genomic abundance. Please note that we have not added these new alternative estimates to the manuscript because we felt they did not add new information.

I don't want to stop them from using the PSMC - it is a fantastic tool - but if they are going to derive their demographic models from single diploid genomes, then the authors should include a discussion of these caveats, at least if they are to rely so heavily on the N_e estimates of the PSMC.

As the reviewer requested, we have added discussion of the limitations of PSMC-derived N_e estimates to the main text. To further address the reviewer's concern, we have also added text describing these caveats to the Supplementary Methods (that we refer to in the in-text discussion) for the reader to reference more detailed explanations of the caveats and potential issues associated with PSMC-derived estimates of N_e . The rationale for these additions to both the main and supplementary text was to try to maintain full transparency about the limitations of PSMC while also maintaining a fairly linear narrative in the main text. In the main text specifically, we have added this section on PSMC caveats as a transition and a justification for also using adult body mass as a surrogate for N_e to further test for evidence of a demographic model. Accordingly, in the revised text, we now make it more clear 'why' we do not solely rely on PSMC-based demographic estimates, and also include estimates based on body mass.

The authors focus on the degree of truncation observed in CR1 and BovB elements across squamate genomes, with the expectation that a greater degree of truncation in the genome would be due to purifying selection. This is an essential assumption of their test of a demographic explanation for variation of repeat content, which they ultimately conclude has "poor explanatory power" for squamate genomes in general. However, their test of the demographic model was critiqued in the previous review, and in my opinion the authors have not adequately addressed this important reviewer concern.

The reviewer is correct in stating that we do incorporate analysis of truncation to test for evidence of a demographic model, but it is not "an essential" aspect of our tests of this model – we also test 4 other features unrelated to truncation. Indeed, truncation represents only 2 of a total of 6 features we test for any evidence of a link with population size:

1. Total TE abundance
2. Total repeat element abundance
3. BovB abundance
4. BovB truncation
5. CR1 abundance
6. CR1 truncation

To address the reviewer's concern and to address any reader confusion about what hypotheses we are testing (and the rationale for these tests), we have re-written a key section of the in-text manuscript to outline the rationale for the features we test for evidence of being correlated with population size (including citations for this rationale) – see response immediately below...

I agree with the author response in that purifying selection COULD act against longer elements and if so COULD result in fewer full-length and more truncated copies - this is predicted by an ectopic recombination model where the strength of selection is related to element length (Petrov et al MBE 2003). But the authors do not explain this at all and their reasoning - if it is along those lines - should be more explicitly stated. Why do

they think more truncation is indicative of purifying selection? Other models exist that explain the nature of purifying selection against TEs (such as gene disruption or toxic transposition), with their own sets of predictions - see Barron et al. Annu Rev Genet 2014. Perhaps their data could be used to test these hypotheses. I won't suggest experiments, but I want to highlight that if the authors want to test the demographic model by using truncation as a measure of purifying selection they need to first establish that purifying selection is (or is not) related to element length.

We thank for reviewer for bringing to our attention that previous versions had failed to clearly outline this rationale. As suggested by the reviewer, in the revised manuscript we have added new text (and citations) to better develop the rationale for using element length as a measure of purifying selection to test the demographic model (see response immediately above). Here, and elsewhere in the revised MS, we are more careful to make distinctions between population-level and phylogenetic-level (i.e., among species) evidence for selection and the role of demography.

From the Revised Results Section (LINE 235) "Multiple studies have suggested that purifying selection acting against TE insertions may manifest in correlations between effective population size (N_e) and features of the genomic TE landscape. This prevailing demographic explanation for variation in repeat content has been invoked to describe patterns of genome complexity and evolution across the tree of life, and predicts that lineages with higher N_e should undergo more effective purifying selection and thus lower genomic accumulation of mutationally hazardous DNA^{40, 41}. Indeed, previous population (within-species) and phylogenetic (among species) studies have provided rationale and empirical evidence that transposable element insertion rates, fixation rates, and abundance may be correlated with effective population size^{14, 41-44}. Relative insert length has also been linked to population size at the population-level by an ectopic recombination model in which element length is correlated with the strength of selection^{14, 18, 42, 45-47}.

Using our phylogenetic-scale dataset, we tested if features of TE landscapes (i.e., genomic abundance, estimated age of activity, and degree truncation for BovB and CR1-L3 LINES) showed evidence of a correlation with estimates of effective population size consistent with a demographic model of TE landscape evolution. We first tested for a relationship between N_e and TE landscape characteristics using the median values of N_e estimates derived from PSMC analyses⁴⁸ for 8 published squamate genomes (Fig. 4b-d, Supplementary Fig. 10). With this dataset, we found no evidence supporting a correlation between N_e and CR1-L3 and BovB length or genomic repeat element abundance (Fig. 4c-d, Supplementary Fig. 10c-e). Notably, we found that species with similar N_e estimates (Fig. 4b) showed different levels of truncation and of TE genomic abundance, and that even within a species TE truncation and abundance were poorly correlated (Fig. 4a, c-d; Supplementary Fig. 10 and 11). Second, to further test for correlations between N_e and element abundance or truncation using an approach that is independent of inferences of generation time and mutation rates, and independent of potential biases associated with coalescence-based estimates of N_e (i.e., population substructure, migration, selection)⁴⁸⁻⁵⁴, we used adult body mass as a proxy for N_e for all species included in our study (as in⁵⁵; Supplementary Table 9,⁵⁶). This approach has the added benefit of leveraging the much larger sample size of our entire dataset (compared to our PSMC analyses using 8 complete genomes). Similar to our PSMC-based analyses, we compared body mass to CR1-L3 and BovB genomic abundance, their degree of truncation, and total genomic repeat element and TE abundances. Consistent with our PSMC-based analyses, we failed to find a correlation between body mass and truncation (Fig. 4e and Supplementary Fig. 12b) that would support a demographic model of TE landscape evolution; the only correlative trend that we did find was a correlative trend that is opposite of that predicted by the demographic model between N_e and genomic repeat element

abundance instead (i.e., higher N_e was positively correlated with TE abundance; Supplementary Fig. 12d). “

Examining a genome assembly by looking at the length and age distributions of truncations can generate the hypothesis that purifying selection may or may not be controlling element diversity - but it still needs to be tested. In the literature, the most convincing tests come from examinations of polymorphism (Petrov et al MBE 2003, Neafsey et al MBE 2004, Blass et al. GBE 2012, Tollis et al. GBE 2013). I understand that the authors only generated short read data from single individuals, but this should not preclude the ability to detect polymorphic insertions using split reads (case in point: they used the same data to detect polymorphisms in single individuals with the PSMC). It was a suggestion made in the previous review, and the authors decided not to apply it. Overall, their test of the demographic explanation lacks power - rather than the demographic explanation itself.

We generally agree with the reviewer that analyses of TE polymorphism in natural populations provide an incredibly powerful approach for understanding selection pressure on TEs and the roles of demography and selection on TE fixation. However, all meaningful and powerful analyses of TE polymorphism require comparisons of the frequency distributions of polymorphic TE insertions to develop estimates of the strength of selection on insert length and other characteristics. Because we only have data from one individual per species, we cannot infer meaningful frequency distributions of polymorphic TE insertions (all we can detect is homozygous vs. heterozygous). This is the reason that we did not conduct these analyses based on previous suggestions, and do not conduct them here.

Importantly, comments from the reviewer made us realize a key point that we failed to clearly point out broadly across our manuscript, and especially in the Discussion – that our study is fundamentally designed to evaluate models of TE evolution at the “phylogenetic” (i.e., fixed differences among-species) level, and not designed in any way to test such hypotheses at the population level (i.e., variation within species). This is an important distinction because the TE literature is comprised of studies conducted at both levels, sometimes with opposing findings when phylogenetic and population-level conclusions are compared. We believe this also represents a significant distinction that lies at the heart of the concerns the reviewer had with the manuscript, that we hope are remedied by our more careful and precise treatment and discussion of these two different types of studies.

We have tried to be far more clear that we are only capable of testing these hypotheses at the phylogenetic scale, given the nature of our dataset (with one individual per species). We have also revised the text to point out that broad conclusions from our phylogenetic-scale study contrast with conclusions from population-level studies on some of the same lineages. We interpret this result as indicating that different modes or models of TE evolution may dominate different evolutionary scales; for example, pre-fixation versus post-fixation processes may be fundamentally different. We also point out that this is a key distinction because there is, indeed, substantial evidence from work on *Anolis* and other species that, at the population level, demography does appear to impact the population frequency of longer TE inserts, which are the major targets of purifying selection.

Neafsey et al (2004) and others have shown that the cause of the deleterious effects of TEs may be very different across taxa, and that certainly may be the case over 200 million years of squamate evolution. The authors invoke this in their discussion and also suggest host regulation may differ between squamate lineages. They provide a tantalizing look into the genome structure of one of the most diverse and enigmatic groups of vertebrates - but ultimately I think the demographic model is inadequately tested here. These shortcomings

need to be addressed - either by more explicitly describing the caveats in their demographic models and their proxy for purifying selection and recalibrating the conclusions in proportion to the inherent weaknesses of those assumptions, or by adding more experiments.

We are pleased that the reviewer appreciates the broader value of the study and study system. Overall, the reviewer suggested that the manuscript should be improved by: “...*either by more explicitly describing the caveats in their demographic models and their proxy for purifying selection and recalibrating the conclusions in proportion to the inherent weaknesses of those assumptions, or by adding more experiments.*”

We have revised the manuscript to address all of these major points of concern brought up by the reviewer in the following ways:

- We have more explicitly described the caveats of our demographic inferences based on PSMC analyses (and provided alternative estimates based on body size as a proxy).
 - We have added key summaries of the major caveats of inferences from PSMC-based *Ne* estimates to the main text, and used this to provide more clear rationale for why we also secondarily used adult body mass as a proxy for population size.
 - We have also added more detailed explanations of the caveats of PSMC-based *Ne* inferences to the Supplementary Methods.
- We have more explicitly described our rationale for using multiple approaches to test for purifying selection, and added new discussion and citations in the main text that justify these approaches
- We have more clearly explained and cited literature to outline the rationale behind our testing of a demographic model (e.g., our use of element length and abundance).
- We have re-written key parts of our rationale and discussion to make far more clear the distinction between phylogenetic-scale and population-scale analyses of TE evolution, and to make the clear point that while our evidence suggests a demographic model is a poor fit to the phylogeny-scale data, other studies at the population level have found evidence for population size being linked to TE features. We have also revised our main conclusions to include that our results together with those from previous studies suggest that different models to explain the primary determinants of TE evolution may dominate different evolutionary scales (e.g., a demographic model may drive population-level phenomenon more strongly than it does at the phylogenetic scale).

Reviewer #2 (Remarks to the Author):

I thank the authors for revising their manuscript by addressing both reviewers' comments. I find these revisions very satisfying and believe that they greatly improved the quality of this work, and thus believe that it should be warranted publication in Nature Communications without additional revisions.

We thank the reviewer for their very thorough and constructive suggestions from the previous round of revision, and are very pleased that our revisions satisfied their concerns.

REVIEWERS' COMMENTS:

Reviewer #1 (Remarks to the Author):

I think the authors have responded accordingly to the critiques and with their patience and perseverance have greatly improved the manuscript! I think the additional analyses have help to exhaust confounding variables to the best extent given the tools used, and the addition of the qualifying discussion points avoid "overselling" while also providing interesting food for thought - particularly the disruptive patterns at the intersection of population-species boundaries.

At this point, I think the paper should be accepted and I look forward to seeing this contribution to squamate genomics in print.